# The Inventory of Nonordinary Experiences (INOE): Evidence of validity in the United States and India

**Ann Taves**[1]*, **Elliott Ihm**[2]*, **Melissa Wolf**[3], **Michael Barlev**[2,4], **Michael Kinsella**[1], **Maharshi Vyas**[1]

**1** Department of Religious Studies, University of California at Santa Barbara, Santa Barbara, California, United States of America, **2** Department of Psychological and Brain Sciences, University of California at Santa Barbara, Santa Barbara, California, United States of America, **3** Gevirtz Graduate School of Education, University of California at Santa Barbara, Santa Barbara, California, United States of America, **4** Department of Psychology, Arizona State University, Tempe, Arizona, United States of America

☯ These authors contributed equally to this work.
* anntaves@ucsb.edu (AT); elliott@ucsb.edu (EI)

**Data Availability Statement:** Validation templates and response data for the final iteration of all validated items have been made publicly available

## Abstract

Researchers increasingly recognize that the mind and culture interact at many levels to constitute our lived experience, yet we know relatively little about the extent to which culture shapes the way people appraise their experiences and the likelihood that a given experience will be reported. Experiences that involve claims regarding deities, extraordinary abilities, and/or psychopathology offer an important site for investigating the interplay of mind and culture at the population level. However, the difficulties inherent in comparing culture-laden experiences, exacerbated by the siloing of research on experiences based on discipline-specific theoretical constructs, have limited our ability to do so. We introduce the Inventory of Nonordinary Experiences (INOE), which allows researchers to compare experiences by separating the phenomenological features from how they are appraised and asking about both. It thereby offers a new means of investigating the potentially universal (etic) and culture-specific (emic) aspects of lived experiences. To ensure that the INOE survey items are understood as intended by English speakers in the US and Hindi speakers in India, and thus can serve as a basis for cross-cultural comparison, we used the Response Process Evaluation (RPE) method to collect evidence of item-level validity. Our inability to validate some items drawn from other surveys suggests that they are capturing a wider range of experiences than researchers intend. Wider use of the RPE method would increase the likelihood that survey results are due to the differences that researchers intend to measure.

## Introduction

Researchers increasingly recognize that the mind and culture interact at many levels to constitute our lived experience [1–4], yet we know relatively little about the extent to which culture shapes the way people appraise their experiences and the likelihood that a given experience

at the Open Science Framework and can be accessed at https://osf.io/w6yhg/ (10.17605/OSF.IO/W6YHG).

**Funding:** This research was supported by John Templeton Foundation (URL: www.templeton.org) grant #61187 (AT). The funders had no role in study design, data collection and analysis, decision to publish, or preparation of the manuscript.

**Competing interests:** The authors have declared that no competing interests exist.

will be reported. Cultural concepts are built into reports of experience and in many cases the categories lay people and researchers use to classify them. For example, according to the General Social Survey about 40% of Americans report having had a religious or spiritual experience *that changed their life* [5]. Yet, this fact tells us little about the phenomenology of the experiences Americans have had, which experiences they consider religious or spiritual, or indeed what the terms "religious" and "spiritual" mean to them. Did they hear a voice telling them what to do and conclude God (or the devil) was speaking? Did they have a feeling of unity or oneness with all things and conclude they had experienced Ultimate Reality? Did they feel a sudden upwelling of joy at the birth of a child and conclude it was an act of divine providence? Did they survive a serious accident and decide it was due to karma? If a person reports having had a transformative experience, they are linking a feature of their experience–a voice that seems to originate outside their head, a feeling of unity or oneness, an upwelling of joy, or surviving an accident–with a cultural concept (e.g., God, the devil, Ultimate Reality, or karma), and they are asserting that the linked concept allows their experience to count as religious or spiritual. The appraisal of the experience, both in terms of this cultural concept and the classification of the experience as religious, spiritual, pathological, etc., are matters of debate and dispute–not only among laypeople, but among religious and spiritual authorities, clinicians, and academic scholars [6–8]. Appraisal, as used here, refers to the outcome of a multi-level process that humans and other animals rely on when determining what is happening, that is, to interpret situations and events [9].

Researchers define constructs, which are typically discipline-specific theoretical ideas, and use them to designate an object of study [10]. For example, researchers in psychology, psychiatry, philosophy, and religious studies define and operationalize constructs, such as religious, mystical, psychopathological, or anomalous experiences, based on observed commonalities among experiences of disciplinary interest. The commonalities that define each set of experiences and distinguish them from one another are not phenomenological features, but culturally-derived disciplinary appraisals of what counts as religious, mystical, psychopathological, or anomalous [11]. These disciplinary classifications tend to obscure the overlap between experiences with similar phenomenological features within a culture and at the same time limit cross-cultural comparisons [12].

Due to the siloing of research on experiences based on discipline-specific constructs, as well as the difficulties inherent in comparing culture-laden experiences, there are few tools that allow researchers to drill down to the phenomenological features of experiences of interest and to compare them across cultures, much less to investigate the ways in which mind and culture interact to shape lived experiences. To overcome these difficulties, we applied a new theoretical approach that separates phenomenological features of experiences from appraisals [13]. Here, we present a new survey instrument–The Inventory of Nonordinary Experiences (INOE)–that queries a wide range of features and appraisals, allowing researchers to make cross-cultural comparisons between English-speakers in the US and Hindi-speakers in India. To ensure that the INOE survey items are understood as intended cross culturally, we present evidence of item level validity using the newly developed Response Process Evaluation (RPE) method [14, 15].

## Why the INOE is needed

When surveys operationalize experiences in terms of constructs such as religious, mystical, or psychopathological, they typically include items that query features of those experiences that are appraised in light of the construct and are thus construct specific. For example, when the Baylor Religion Survey asks people if they "personally had a vision of a religious figure while

awake," "heard the voice of God speaking to [them]," or "felt called by God to do something," it links features–seeing visions, hearing voices, and an inner compulsion to act–with God or religion more generally. Similarly, when the Baylor Religion Survey asks people if they have "spoke[n] or prayed in tongues" or "received a miraculous, physical healing," it links automatic speech with experiences described in the New Testament in the first case and healing with the concept of miracles in the second. The phenomenological features that the Baylor Religion Survey casts in religious terms, however, also appear in surveys designed to measure experiences researchers consider pathological and/or paranormal. Thus, visual or auditory hallucinations and the feeling of being compelled to a particular course of action can be viewed as psychopathological; automatic movements and healings can also be attributed to psychic or paranormal powers.

In an attempt to avoid both religious and pathological connotations, researchers have introduced more expansive constructs. Some, whose interests arose in response to the effects of psychedelic drug use, refer to altered states of consciousness. Others, whose interests descend from the psychical researchers and parapsychologists of the 20th century, have advanced constructs such as extraordinary, anomalous, or exceptional experiences. In doing so, they also rely on binaries such as ordinary versus extraordinary and everyday versus uncommon [16–19]. However, because culture-specific practices and expectations, both overt and tacit, have the potential to make experiences seem more or less ordinary and, thus, to shape what people attend to, remember, and recount, we cannot assume that these binaries are cross-culturally stable; in other words, we cannot assume that what counts as extraordinary, anomalous, or exceptional is the same across cultures [20]. In response to a given culture or way of life, people may single out particular experiences as worthy of special attention, cultivate valued experiences, and discount or even disparage experiences valued by others [13]. At the same time, some experiences that are common in a population (e.g., marriage, the birth of a child, the death of a loved one, and accidents) may stand out for people as significant and potentially life changing regardless of their culture.

There are currently many existing scales designed to measure experiences based on researcher-defined constructs, such as psychopathological, religious, spiritual, mystical, or anomalous. However, these scales often contain similar items. Although researchers generally recognize this overlap, they have dealt with it by adding caveats to their definitions. Psychiatrists are interested in dissociative and hallucinatory experiences that cause distress and are not "a normal part of a broadly accepted cultural or religious practice" [21]; anomalistic psychologists are interested in experiences that "deviate . . . from the usually accepted explanations of reality according to Western mainstream science" [16]; philosophers of religion are interested in experiences that might point to transcendent realities [e.g., 22]. However, when researchers use caveats to narrow the focus of their research, they exclude from consideration many experiences that have similar phenomenological features. For example, a clinical researcher interested in dissociative identity disorder may ignore experiences of spirit possession that do not meet clinical criteria (e.g., because they are an accepted cultural or religious practice), even though such experiences may be relevant for understanding the neuropsychological and cultural processes underlying dissociation.

The use of these caveats to distinguish researcher-defined types of experiences limits our ability to study the processes that shape and differentiate lived experiences. This runs two risks. First, because they don't look at the full range of experiences that contain the features of interest to them, researchers risk reifying their own parochial, culturally-based constructs and imposing them in contexts where the feature(s) of interest are interpreted differently. Second, it inhibits our ability to investigate the role of appraisal processes (and culture more broadly) in shaping what people experience and how it affects their lives. Thus, we don't know the

extent to which valorizing and cultivating experiences increases the likelihood that experiences with certain features will be reported. Nor do we know to what extent distress is linked to pathological appraisals in the absence of cultural practices and expectations that might normalize an experience. Answering these questions requires us to set aside discipline-specific theoretical constructs and take a feature-based approach to the comparative study of experiences.

## The design of the INOE

The Inventory of Nonordinary Experiences (INOE) has two key design features: it takes a feature-based approach to comparing experiences and it adopts a subject-dependent (emic) definition of "nonordinary."

**A feature-based approach to comparing experiences.** In order to compare experiences across cultures, the INOE separates phenomenological features of experiences from their appraisal, and queries both. Here, we use 'phenomenological features' of an experience to refer to generically describable aspects of what it was like or seemed like as it occurred and 'appraisals' to refer to the meaning or significance that is ascribed to the experience as it occurs or after the fact. We refer to this as a "feature-based" approach [13]. It was inspired by earlier efforts to distinguish between the phenomenological features of experiences and their appraisals in interviewing protocols [23] and to a more limited extent in surveys [18, 24]. The feature-based approach allows us to separate potentially universal (etic) and culture-specific (emic) aspects of lived experiences in a new way, and it treats a phenomenological feature of a lived experience as the basis for comparison.

However, in doing this we run the risk of imposing features that seem universal to us on other cultures. To overcome this risk, which has long been recognized by cross cultural psychologists [25], we needed to ensure that the description of the common feature was a *derived etic* (i.e., a formulation that makes sense in the cultures to be compared) and not an *imposed etic* (i.e., a formulation derived from one culture and imposed on another). We adapted Berry's [25] ethnographic method for identifying derived etics by dividing it into two steps.

In the first step, we separated *experience items*, which asked people whether they had had an experience with a certain feature, from *follow-up items*, which asked those who had had the experience how they appraised it. In doing so, we separated what we refer to as *proposed etics*, that is, items with features that we thought people would likely recognize across cultures, from additional aspects of their lived experience, including their *emic* appraisals of what happened and why.

Then, in the second step, we used a newly-developed item validation process–the RPE method–to test whether a proposed etic was actually understood as intended in the cultures to be compared and thus could be considered a *derived etic*. The RPE method, which turns cognitive interviews into open-ended meta-surveys through the use of web probes, allowed us to assess whether respondents understood items as intended in both cultural contexts and, if not, to iteratively revise and re-test them [14, 15, 26]. In contrast to Berry's approach in which researchers identify the common feature based on their understanding of the cultures involved, our approach relies on the ability of people from within each culture to recognize an (etic) common feature in their own lived (emic) experience. Using the RPE method, we were thus able to test whether subjects could recognize the generically-worded experience items in their lived experience and whether, when participants said they had had an experience, they had a specific lived experience (or in a few cases, a type of experience, as discussed in the Cautions section) in mind that allowed them to answer the follow-up questions.

**A subject-dependent definition of "nonordinary".** Cross-cultural psychologists have developed ways to combine presumably universal (etic) and culture-specific (emic) approaches

to constructs of interest [27], as well as methods for minimizing bias with respect to constructs, methods, and items when making cross-cultural comparisons (for an overview, see [28]). These discussions generally assume that researchers want to measure an overall construct, such as "personality" [29], "paternal warmth" [30, 31], or "filial piety" [32].

Because we cannot assume that the ordinary-nonordinary distinction is cross-culturally stable [13, 20], we adopt a subject-dependent (i.e., emic) definition in which "nonordinary" refers to experiences that stand out to people or are marked by them as special relative to what *they* consider ordinary or everyday. In light of this shift to an emic definition of nonordinary, which holds open the possibility that different experiences might stand out for people in different cultures, it is not meaningful to try to measure an overall (researcher-defined) theoretical construct. The INOE, as a result, is designed to measure distinct experiences, stripped down to their phenomenology, without presupposing whether or how they cluster across cultures. With this design, each item in the INOE is essentially its own construct; there is no overall construct that needs to be measured and, as a result, no predefined set of phenomenological features that needs to be queried in the INOE. In terms of the design of the INOE, this means that we needed to justify the inclusion of items on other grounds.

The items included in the INOE meet several criteria. First, they are subject to conflicting interpretations. Second, the interpretations often involve outsized claims having to do with intervention by deities or other supernatural beings or forces, extraordinary abilities, and/or psychopathology. Third, they often have profound effects on people's lives—whether positive or negative. Given these criteria, the INOE not only includes experiences that commonly appear on instruments designed to measure various researcher-defined conceptions of "nonordinary," but also a wide range of experiences that are marked as special (and, thus, valorized) and/or cultivated by different religious and spiritual traditions around the world. We recognize that this set of criteria does not define a clearly delimited set of experiences. We do not claim that the items included in the INOE are exhaustive–nor do we aim for them to be. In terms of use, this means that researchers and clinicians do not have to administer the INOE as a whole, but can select items that are most appropriate for their uses. In the remainder of the Introduction, we elaborate on the way we constructed and validated items.

## Item construction

The INOE is made up of experience items and follow-up items. Many of the experience items were adapted from existing scales and others were created to capture experiences valorized and cultivated by various traditions. The follow-up items were designed to capture the context, the effects of the experience, and the way that people categorized and explained it. For ease of discussion and the convenience of researchers, we gave the items nicknames and grouped items under headings, e.g., Emotions, Presences, Meaning.

**Experience items.** We culled items from existing scales, refining the wording as needed to focus on the phenomenology of experiences, and, in so far as possible, stripping the items of appraisals. Many of these items were uncommon bodily and sensory experiences that psychiatrists characterize as hallucinations, dissociations, and alterations in sense of self and others as visions, voices, spirit possession, and mystical experiences. To these, we added experiences–some of them quite common–that we knew were marked as special, and in some cases, valorized and cultivated by religious and spiritual traditions. We also created items that query types of things—i.e., objects, places, and persons—that people might mark as special. Because many cultural traditions valorize and cultivate positive emotions and attempt to mitigate the effects of negative emotions, we added emotion items as well. People often associate these items with life events (e.g., childbirth, marriage, accidents, or the death of a loved one) that stand out for them.

**Table 1. Involuntary movement items in scales measuring discipline-specific theoretical constructs.**

| Item | Source | Discipline-specific Construct |
|---|---|---|
| "I have **spoken in tongues**." | Anomalous Experience Inventory [33] | Anomalous experiences |
| "Have you ever had an experience in which you felt your body **moving automatically**, or felt urges to move into certain postures or make certain movements, when you didn't seem to be controlling this?" | Appraisals of Anomalous Experiences Interview Probes [AANEX; 23] | Psychotic-like experiences |
| "Have you ever thought that other people or agencies were **putting thoughts in your head**, or making you feel certain things?" | AANEX | Psychotic-like experiences |
| "When I sing or write something, I sometimes have the feeling that **someone or something outside myself directs me**." | Creative Experiences Questionnaire [CEQ; 34] | Fantasy proneness |
| "I have experienced **my body or parts thereof**, such as my limbs or my voice, **moving or operating automatically** and without my conscious intervention." (trans. from German) | Questionnaire for Assessing the Phenomenology of Exceptional Experiences [PAGE-R; 18] | Exceptional experiences |
| I have had an experience in which it **seemed like my body was performing actions outside my control** (such as moving, speaking, or writing). | INOE—*Automaticity* | None |

References to involuntary movements appear in bold. Adapted from [35].

*How we adapted items from existing scales*. When reviewing existing scales, we found many phenomenologically similar items on scales that were designed to measure different constructs. For example, items related to involuntary movement appear on scales oriented toward anomalous, pathological, and creative experiences. See Table 1.

When considering these existing scale items (rows 1–5 in Table 1), we recognized that the phenomenological experiences they described were not identical. Whereas some focused explicitly on seemingly involuntary experiences (rows 2 and 5), others did so only implicitly (rows 1, 3, and 4); some highlighted the sense of an external agent (3, 4); and some included or implied a context (1, 4). In formulating our *Automaticity* item, we focused on the involuntary experience ("my body was performing actions outside my control") and added a range of examples ("such as moving, speaking, or writing"). In contrast to most of the items reviewed, we inserted "it seemed like" to encourage respondents to say "yes" based on what the experience felt like rather than on what they thought was happening (at the time or upon later reflection). We deliberately left the sense of an external agent out of this formulation as it is the focus of separate items on the INOE.

Items that refer to nonphysical presences also appear in a wide range of scales, where they refer to anything from God's presence to evil presences, ghosts, aliens, and even magical beings such as elves and fairies (see Table 2).

All the items in this table, including our own, specified the kind of presence that was of interest. The appraisals in the first five items specified a *specific* type of presence that was relevant to the construct being measured in the scale, e.g., "God" in a spirituality scale and an "evil presence" in a psychosis scale. The PAGE-R and INOE both use generic terms to refer to presences (i.e., as a force or entity), which then had to be qualified to distinguish them from physical presences. The PAGE-R refers to presences "invisible to the *ordinary* senses" (emphasis added); we refer to "non-ordinary" presences. (For the sake of clarity, we hyphenate 'non-ordinary' in the INOE items.)

Finally, as with the *Automaticity* item, we inserted "seemed to be" to capture what experiences of presence felt like regardless of whether it was appraised as real or not. The inclusion of "seemed to be" in the English version encouraged participants to interpret the item based on how the experience felt to them, without committing them to saying that such a presence actually existed. The Hindi equivalent–*anubhav*–inherently stresses how the experience seemed to be.

**Table 2. Items about sensed presences in different scales.**

| Item | Source |
|------|--------|
| "I feel **God's presence**." | Daily Spiritual Experiences Scale [36] |
| "Did you seem to encounter **a mystical being or presence**?" | Near Death Experiences Scale [NDES; 37] |
| "Did you see deceased **spirits or religious** figures?" | NDES |
| "I have seen a **ghost or apparition**. . . . an **extraterrestrial** . . . **elves, fairies**, and other types of little people." | Anomalous Experience Inventory [33] |
| "Have you sometimes sensed an **evil presence** around you, even though you could not see it?" | The Oxford-Liverpool Inventory of Feelings and Experiences [O-LIFE; 38] |
| "I have felt the **presence of a force, energy, entity, or atmosphere** that is invisible to the ordinary senses." (trans. from German) | Questionnaire for Assessing the Phenomenology of Exceptional Experiences [PAGE-R; 18] |
| "I have sensed **the presence of what seemed to be non-ordinary forces or entities**." | INOE—*Presence (nonordinary)* |

The references to sensed presences appear in bold. PAGE-R did the most to strip its item of cultural framings.

*How we developed new items*. In addition to items adapted from existing scales, we constructed items to capture experiences we knew were valued and cultivated by particular religious or spiritual traditions (see Table 3). For example, to capture an experience some would view as transcendent: "I have felt small in relation to something vast or powerful." Inspired by reports of seeing the face or form of Jesus and Hindu deities in objects, we added "I have seen what seemed like a face in a natural or human-made object." Inspired by the Hindu practice of *darshan*, we added: "I have seen and/or interacted with images, statues, or other physical objects that seemed to contain a non-ordinary presence or power."

In constructing new items, we used qualifiers as needed to distinguish subsets of experiences that would otherwise be relatively commonplace. All the emotions items are qualified by a stem that specifies that the experience "stood out from all other such experiences." For example: "I have had an experience of joy, ecstasy, or bliss *that stood out from all other such* experiences."

**Follow-up items.**   The follow-up items allow us to explore subjects' lived experiences, which we discuss under four headings: Mental States, Effects, Categorization, and Cause.

*Mental state item*. The mental state query allows us to investigate whether the experience occurred under normal waking conditions or under a range of other conditions, such as during sleep or in the interval between sleeping and waking, under the influence of psychoactive substances, or while mentally or physically exhausted. Many existing surveys do not include

**Table 3. Inspiration for some experience items.**

| Category | Item nicknames | Inspiration |
|----------|----------------|-------------|
| **Emotion** | Pain, Loss, Pleasure, Misfortune<br>Awe, Compassion<br>Places (special)<br>Devotion (objects)<br>Devotion (people) | Emotions potentially moderated by appraisals<br>Secular spiritualities; Buddhism<br>Holy places; Indigenous traditions<br>Cherished and sacred objects<br>Saints, gurus, & charismatic leaders |
| **Sensory** | Faces | Jesus in the tortilla; Ganesh in the tree |
| **Sense of Self** | Absorbed<br>Diminished Self | Association with visualization & meditation<br>Experiences of transcendence |
| **Presences** | Places (animated)<br>Objects (animated) | Indigenous traditions<br>Hinduism |

experiences that take place in these "altered states," because they are focused on psychopathology or occurrences that might challenge conventional science. However, many cultural traditions value these altered states, viewing dreams and/or psychoactive drugs as means of interacting with divine entities, acquiring transcendent knowledge, and/or entering spiritual realms. In light of our interest in the effects of culture on experience, querying a wide range of mental states is crucial.

*Mental State*: "When you had the experience, were you . . . (Select the most important)."

(Using drugs or alcohol; Affected by mental or physical illness; Falling asleep, waking up, or exhausted; Asleep (dreaming); None of the above)

*Effect items*. We designed additional queries in order to determine the significance of the experience, its impact on the individual's life, and whether the overall effect was positive or negative.

*Impact*: "Overall, how much of an impact has this experience had on your life?" (Little or no impact; Some impact; Major impact)

*Life Effect*: "Overall, has the **lasting** effect of this experience, on your life or beliefs, been more positive or negative?" (Very positive effect; Somewhat positive effect; Neutral or no effect; Somewhat negative effect; Very negative effect)

*Category (R/S)—Religious/Dharmik or Spiritual/Adhyatmik*. Because we are interested in whether participants categorize their experiences as religious or spiritual, we ask, "Do you consider this experience religious or spiritual?" Although many in the U.S. and Europe distinguish between "religious" and "spiritual", these terms are used in overlapping and inconsistent ways [39]. We thus refer to "religious or spiritual" and gather more precise information regarding participants' religious and spiritual identity and practice in the demographics. When we translated this item into Hindi, we used *dharmik*/धार्मिक in place of religious and *adhyatmik*/आध्यात्मिक in place of spiritual, recognizing that these terms are not necessarily equivalent.

Although we want respondents to understand the experience items in the way we intend in each culture in which we administer the INOE, the meaning of key concepts in the follow-up items can vary cross-culturally as long as we know how the term is understood by respondents and we interpret the results in light of their understanding. This means that researchers can use a culturally-inflected concept, such as "religion," in those contexts where it is culturally appropriate and incorporate other key concepts, such as *dharma* (Sanskrit) in India or *dao* in China. Whatever terms are used, it is incumbent on researchers to understand the meaning(s) of the terms within each culture for respondents and interpret respondents' responses in light of their understanding. Because scholars of South Asian religions stress the complexity of the concept of 'dharma' and resist equating it with the Western concept of 'religion' [40, 41], we were aware that in validating this follow-up item we would need to determine how respondents understood religious/dharmik and spiritual/adhyatmik, rather than assuming a uniform understanding across US and Indian populations.

*Category (R/S)*: "Do you consider this experience spiritual or religious?" (Yes, No).

*Cause items*. Recognizing that some view religion/spirituality and science as competing causal explanations and others as compatible [42], we included the Science query. Because there is evidence that people simultaneously use science to explain how something happens and supernatural beliefs to explain why something happens [43], we included the Reason query to investigate the extent to which people who think science can explain their experience (i.e., say 'yes' to the Science query) and the extent to which even those who do not consider the experience religious or spiritual (i.e., say 'no' to the Category (R/S) query) nonetheless offer nonscientific reasons for why something happened to them. Finally, recognizing that not all forms of religion or spirituality involve unseen agents [44], we ask if they think spiritual beings or forces were involved.

*Science*: "Do you think science can explain how this experience happened?" (Yes, science can or will be able to explain it; No, something More is involved.)

*Reason*: "Why do you think it happened to you? (Select the closest answer.)" (To offer me a sign or message; To reward or punish me for my actions; Due to destiny/fate; None of the above [may include chance/probability])

*Agent*: "Who, if anyone, caused you to experience this? (Select the most important.)" (God or gods; Other spiritual beings or forces (including the dead); None of the above)

## Item validation

In the social sciences, validity broadly refers to the appropriateness of the use of survey results (or other type of assessment) for an intended purpose [45, 46]. As such, the type of validity evidence that should be presented depends upon the intended use of the survey and the intended interpretation of the survey responses [47, 48]. Our goal with the INOE was to develop an inventory of discrete generically-worded experiences, along with follow-up items, that could be used to investigate the experiences reported by English-speakers in the US and Hindi-speakers in India. We chose these two populations because they are both historically dominated by distinctly different religious cultures (Hinduism in India and Christianity in the US) that valorized different sorts of experiences and because both contain diverse subpopulations that we hypothesized might be affected by their respective dominant cultures. We translated the INOE into Hindi, rather than simply administering it in English in India, because we wanted to test whether we could establish a basis for comparison despite the deep connections between language, culture, and experience. In light of our goal, we therefore needed to pretest items and collect response process evidence to ensure items were understood as intended within and across cultures [49–51]. This can be thought of as establishing item level validity [14, 15].

Given our aims, we could not rely on the kind of evidence of validity that is traditionally presented in psychology, i.e., correlational evidence for validity based on the internal structure of a construct or relationships to other variables. This is because we did not design groups of items to measure one or several constructs; rather, each item is essentially its own construct. As such, quantitative approaches to validating instruments for use in latent variable models (e.g., factor analysis, item response theory) are not theoretically consistent with our item level approach. Instead, to validate our instrument we needed to follow each participant's response process to collect evidence that each item was understood as we intended in both cultures (with the exception of the Category (R/S) follow-up item (religious/spiritual and dharmik/adhyatmik), which we deliberately translated in culture-specific terms, as discussed below). Participants' demonstrated comprehension, or lack thereof, allowed us to identify which of our items (proposed etics) were well understood (and thus could be considered as derived etics) and revise or eliminate items that were not well understood. Items that were consistently understood as intended by a large proportion of participants within each culture are considered to be "validated," where validated means that the inferences drawn from the item responses can be trusted and used to draw comparisons within and across these cultures. It is worth noting that validation is an on-going process and new evidence could arise that no longer supports the use of these items, especially as shared meanings with and across cultures evolve over time [48].

The item-level validity evidence we needed is described by the *Standards for Educational and Psychological Testing* as evidence based on the response process [47]. The response process is the cognitive process that a person engages in when responding to an item [52]. This may include subconscious or conscious activities such as reading the item, interpreting it within a

particular cultural framework, and formulating or selecting a response [53, 54]. However, the response process is rarely investigated [55–57]. This is due in part to how much time it takes to collect and analyze qualitative data, as well as to document improvements in interpretation, especially across multiple cultures [58, 59].

When researchers do investigate the response process, they typically use cognitive interviews or focus groups, in which participants are instructed to 'think-aloud' and verbally describe their thought process as they respond to an item [49], or web-probing, which turns cognitive interviews into meta-surveys, in which participants respond to open-ended questions about each item [60, 61]. Although web-probing has streamlined the collection of some response process data, neither cognitive interviews nor web probing are designed to test and re-test revised versions of items, in an iterative fashion, to determine if revisions have made the item clearer or reduced misunderstanding. In addition, neither approach provides a framework for reporting important elements of item-level validity.

To overcome these limitations, we used the Response Process Evaluation (RPE) method [14, 15], which provides a way to iteratively test if revisions improve the interpretability of an item both within and across cultures. As a form of web-probing, the RPE method uses meta-surveys consisting of several *probes* about survey items that prompt participants to explain two parts of their response process: their interpretation of the survey item and their rationale for selecting a response option. When completing the meta-survey, participants respond to *probes* about the survey *items*. The RPE method uses iterative meta-surveys to reach a version of each item that is understood as intended, within a single population or across multiple populations simultaneously. It also introduces a framework for documenting important elements of item-level validity: the intended interpretation of each item, the range of participants' interpretations, and the proportion of respondents who understood each item as intended. The end result is a series of item validation reports that document the item validation process.

The RPE method allowed us to test the two key assumptions built into the INOE: first, that respondents could recognize the features in their lived experience (or give a hypothetical example if they have not had this experience), and, second, that when respondents say "Yes" to an experience item, they are responding in light of a sufficiently *specific* experience that would allow them to answer follow-up items, rather than a vague sense of having had such an experience (for discussion of exceptions, see Cautions section below).

## Method

### Translation

Three native Hindi speakers independently translated the INOE from English to Hindi, keeping in mind the intended interpretations of the items. Two versions were created: one in Devanagari script (e.g., "मैंने असाधारण शक्तिओं या वस्तुओ कि उपस्थिति को महसूस किया है।") and one in Roman script (e.g., "Maine asadharan shaktiyo ya vastuo ki upasthiti ko mehsus kiya hai"). A fourth native Hindi speaker resolved discrepancies between translations from each of the three translators, in conversation with the translators and team members. This Hindi translation was then back-translated to English by two additional native Hindi speakers who were naive to the original English items. The two back-translations were reviewed and minor discrepancies were resolved. Indian participants were given a choice between the Roman and Devanagari versions at the beginning of each survey.

### Item validation (RPE method)

**Participants.** Participants were recruited on Amazon's Mechanical Turk [MTurk; 62] via the CloudResearch platform, using exclusion criteria of a 98% approval rating and 1,000 prior

studies completed. The probes for experience items and follow-up items were responded to by, respectively, 1144 and 357 Hindi speaking Indians and 1080 and 489 English speaking Americans. US participants identified as 38.1% female 61.0% male (1.0% other), with a mean age of 35.9 (SD = 10.9). (Demographic data for 200 participants who responded to probes for follow-up items are missing due to a programming error. Descriptive statistics are given for the remaining 1,369 participants.) Indian participants identified as 22.5% female and 77.5% male, with a mean age of 35.4 (SD = 9.57). We used screener items at the beginning of the survey to reduce the prevalence of low-quality respondents, or "farmers" [63], and disqualified all participants that made it past these screener items but gave unintelligible responses to our open-ended survey items. Participants were paid minimum wage ($8.00/hr in the U.S. and $6.25/hr in India) proportional to the average amount of time expected to participate in the study [64]. Ethical approval for this study, which was limited to persons over 18, was obtained from the UC Santa Barbara Human Subjects Committee, who waived the need for written consent (Protocol #18-22-0723).

**Materials and procedures.** Informed consent was obtained using an online information sheet. Participants then responded to items in batches of roughly five participants each, and items were iteratively revised until 20 participants had evaluated the final version of each item.

*Probes.* Our probes queried each participant's *interpretation* of each INOE survey item and the rationale behind the *response* option they selected. Taken together, these comprise the participant's overall *response process*. Probes are used in cognitive interviewing and web probing to pretest items and response options for interpretability and clarity [60, 61, 65, 66]. For example, participants might be asked to restate the survey item in their own words (paraphrase probe), define a key word or phrase (comprehension/interpretation probe), or explain why they selected a particular response option (category-selection probe).

The probes followed the same broad pattern for each survey item, with slight variation depending on the item. Since there is no single correct way to write a probe, a probe was considered correctly written when it elicited the information necessary to determine if the item was understood as intended. The INOE experience items were typically queried using three or four probes and focused on the survey item itself, rather than the response options, because the substance of the experience items was generally found in the survey item. The INOE follow-up items were queried using anywhere between two and eight probes and tended to focus on both the survey item and corresponding response options, because the substance of the follow-up items was found in both. We consider an example of each item type below.

**Experience items.** To demonstrate the probes for the experience items, we will use U.S. responses to our *Faces* item. The first iteration of this item was "I have seen a face in a natural or human-made object." Participants were shown the item and asked to respond to a series of probes (see Table 4; [56, 62]). The *response probe* asked participants how they would respond to the item (i.e., "Yes" or "No"). The *paraphrase probe* asked participants to paraphrase the item in their own words. For some items, participants also answered a *comprehension probe*

**Table 4. Example of probes for experience items: *Faces*.**

| Probe | Probe Type | Response Type | Response Options | Participants |
|---|---|---|---|---|
| 1. If these were the response options, which would you select? | Response | Closed | Yes/No | All |
| 2. In your own words, what does "seeing a face in a natural or human-made object" mean to you? | Paraphrase | Open | | All |
| 3. Briefly describe your experience of seeing a face in a natural or human-made object. | Example ["Yes"] | Open | | Selected "Yes" on first probe |
| 4. Please give an example of what such an experience would be like, even though you have not had one. | Example ["No"] | Open | | Selected "No" on first probe |

that asked them to define a key word or phrase in the item, e.g., "compassion". Finally, an *example probe* asked participants to give an example of the experience described by the item: if a participant said they had had the experience (i.e., selected "Yes"), they were asked to describe it; if they said they had not (i.e., selected "No"), they were instead asked to give a hypothetical example. Examples were important because they told us what specific experiences participants in a given culture had in mind when paraphrasing the experience item–this allowed us to establish whether they understood the experience in the way we intended and could recognize it in their culture. We also asked participants if they felt they understood the item and included a catch-all probe asking if they had any additional feedback. These rarely yielded any valuable information and are therefore not discussed here.

The response probe was presented first in order to simulate the experience of responding to each item when taking the INOE itself. The open-ended paraphrase and category-selection probes were designed to elicit more thought and reflection than the response probe and were therefore administered only after the response probe, since they may have influenced the response probe had they been asked first [26, 60].

**Follow-up items.** Given the design of the INOE, the follow-up items must be answered in reference to a specific experience item to which a participant has responded "Yes." Therefore, to query the follow-up items, we first presented participants with a set of experience items. If they responded "Yes" to any experience items, they were asked to describe one of their experiences, and then to consider one or more follow-up items in reference to that experience.

Participants were first given a response probe, followed by a category-selection probe, asking them to explain why they selected that response option. The remaining probes for the follow-up items were slightly different for each item because each had different response options. For example, for items with ordinal response options (e.g., *Life Effect*), participants were also asked to explain why they selected one option over an adjacent option (category-rejection probe). This allowed us to test if the magnitude of the Likert scale response options was being interpreted consistently across individuals. For the Agent item, participants were asked to give an open-ended response to the prompt before selecting a response option (open-response probe), in order to see what categories would emerge organically while we revised the response options. Finally, for all items, participants were given a catch-all probe, asking if any response options were missing. We did this to ensure that our response options were mutually exclusive and exhaustive, and to check that we were not imposing our own biases on the items by mistakenly omitting response options that best captured the participants' lived experiences. See Table 5 for an example, in English, of the probes for the *Life Effect* follow-up item. The final version of the *Life Effect* item asked "Overall, has the lasting effect of this experience, on your life or beliefs, been more positive or negative?"

Because the Category (R/S) item included cultural concepts that might not be understood in the same way across cultures, our goal was to determine how respondents understood the four key words (religious/dharmik and spiritual/adhyatmik), rather than seeking to achieve a common understanding across US and Indian populations. In order to assess this, we gave respondents two interpretation probes, one for each keyword, in addition to the response and category selection probes.

**Intended interpretations.** In order to create useful probes and improve consistency in coding participant responses across coders, we developed *intended interpretations* for each item. Intended interpretations state the way that we, the researchers, wanted participants to understand each item, given how we intended these data to be interpreted and used. They help ensure that coders share a common understanding of what is intended when evaluating participant responses (discussed more in the next section) and make it possible for other researchers to borrow items for use in other studies.

**Table 5. Example of probes for follow-up items: *Life effect*.**

| Probe | Probe Type | Response Type | Response Options | Participants |
|---|---|---|---|---|
| 1. Overall, has the lasting effect of this experience, on your life or beliefs, been more positive or negative? | Response | Closed | Very positive effect Somewhat positive effect Neutral or no effect Somewhat negative effect Very negative effect | All |
| 2. Please explain why you selected this response. | Category-Selection | Open | | All |
| 3. Why did you select 'very' instead of 'somewhat'? | Category-Rejection ["Very'] | Open | | Selected "very" on first probe |
| 4. Why did you select 'somewhat' instead of 'very'? | Category-Rejection ["Somewhat"] | Open | | Selected "somewhat" on first probe |
| 5. Do you think any response options are missing? If so, what? | Catch-All | Open | | All |

Our intended interpretations for the INOE experience items differ in their complexity. Some, such as *Faces*, offer a basic dictionary definition and a few examples. Others, such as *Presence (Non-ordinary)*, are more complex because we wanted to include a variety of named experiences that involve the feature and clarify the relationship between this item and others to which it is related or with which it might be confused. See Table 6. Relationships are indicated in "Related Items" and distinctions in "Differential Diagnosis." See S1 Appendix for the list of intended interpretations for all items.

We deliberately did not attempt to develop an intended interpretation for the Category (R/S) follow-up item, because for this question, our goal was not to establish that the terms "religious" and "spiritual" (or "dharmik" and "adhyatmik") were understood consistently across participants. Rather, we wanted to know if participants would use the term(s) to describe their experience.

*Evaluation.* English-speaking coders and bilingual Hindi-English-speaking coders concurrently coded item responses from the US and India, respectively. Prior to weekly team meetings, coders independently evaluated each participant's responses for each item in light of its intended interpretation, taking into account the participant's responses to all probes about the item. Each participant's responses for each item were rated by at least two coders on a 5-point scale: 'Understood' (1), 'Probably understood' (2), 'Not enough information' (3), 'Probably not

**Table 6. Intended interpretations for two experience items: *Faces* and *presence* (non-ordinary).**

**Faces.** "I have seen what seemed like a face in a natural or human-made object."

Intended Interpretation: Seeing faces in everyday objects. Example: Seeing a face in a U.S. electrical socket, the face of Jesus in the tortilla, etc. In psychology, this is referred to as pareidolia.

**Presence (Non-ordinary).** "I have sensed the presence of what seemed to be non-ordinary forces or beings."

Intended Interpretation: A sense of a nonphysical "other," whether perceived as an agent (e.g., deity, spirit, ancestor, dead person, alien, or alter personality) or a more amorphous power, force, or energy, that seems present based on some sort of perceived internal or external cues or messages.

Note: Although alter personalities can be construed as having a body, they are included as long as they seem to have a "mind of their own." Beings, such as tulpas (Veissiere, 2016), that have been cultivated are fine. The perceived presence can be positive or negative, sought or unsought, welcome or unwelcome. The nonphysical power or agent can be disembodied or thought to manifest through a living being or material object. The power or being may simply <u>seem</u> to be present; whether the subject actually thinks it is present will come out in the appraisals.

Differential diagnosis: Signs or messages without the sense of an "other" being present fall under our **Messages** item.

Related Items: This is the most inclusive of our Presence-related items. The others are all more specific.

understood' (4), or 'Not understood' (5). A rating of 1 or 5 indicated that a coder was confident that a participant's responses to an item's probes were consistent (1) or inconsistent (5) with the intended interpretation of the item (i.e., 'beyond a reasonable doubt'). A rating of 2 or 4 indicated that the response appeared to be consistent (2) or inconsistent (4) with the intended interpretation, but more information would be required to confirm this (i.e., 'preponderance of evidence'). A rating of 3 indicated that the response did not contain enough information to evaluate whether the item was interpreted as intended. If a response did not address the probe prompts at all (e.g., low-quality or automated responses; [59]), a rating of 3* was given.

During weekly team meetings, English and Hindi coders discussed their ratings. Responses that were rated 1 or 2 by all coders were given an overall rating of 'Understood' (U), those that were rated 4 or 5 by all coders were given an overall rating of 'Not Understood' (NU), and those that were rated 3 or 3* by all coders retained that score as their overall rating. If coders disagreed about how to code a participant's response, the research team discussed the coding of the response until a consensus rating was reached. If problems were identified with an item, the item was revised as needed, relaunched in small 'batches' of five to ten participants, and the new responses were rated by coders. This process of revising, relaunching, and rating was repeated iteratively until a version of the item was evaluated by at least 20 participants, of whom at least 80% understood the item as intended. Additional responses were collected for follow-up items with one or more response options that were rarely endorsed.

We recorded the proportion of users that understood each item as intended, i.e., the total proportion of responses with an overall rating of 'Understood', excluding 3s and 3*s (Proportion Understood $= \frac{U}{U+NU}$). The final iteration of each item needed to be understood as intended at least 85% of the time in both cultures to be included in the INOE. Items that were not understood as intended at least 85% of the time in one or both cultures were removed.

An experience item with a high Proportion Understood may still have a high rate of false positive responses (saying 'Yes' when they have not had the experience) or false negative responses (saying 'No' when they have had the experience). This can occur when two conditions are both met: (1) responses are substantially skewed toward one response option, and (2) a large proportion of NUs are concentrated among the other, rarely endorsed response option (e.g., only two participants said "Yes" to an experience item and one of their responses was rated as NU). To identify such cases, we computed the proportion of 'Yes' responses that were rated as 'Understood' (Positive Proportion Understood; $NPU = \frac{U_{No}}{U_{No}+NU_{No}}$), and the proportion of 'No' responses that were rated as 'Understood' (Negative Proportion Understood; $NPU = \frac{U_{No}}{U_{No}+NU_{No}}$). PPU and NPU constitute evidence of test-criterion validity [47]. They roughly correspond to PPV (Positive Predictive Value) and NPV (Negative Predictive Value) in the epidemiological literature [67]. Items rated as Understood by at least 85% of participants in both the US and India, but with PPU or NPU values below 80% in either the US or India, were noted. We leave it to other researchers to decide in each case whether the item is fit to use for their research purposes based on the validation results for the experience items (see below).

Similarly, a follow-up item may have a high Proportion Understood despite one response option being poorly understood. A separate Proportion Understood was computed for each response option, and those for which we failed to reach the 85% Understood threshold were noted.

***Evaluation of the Category (R/S) follow-up item*****:** In light of our interest in determining what the terms 'religious'/'spiritual' and 'dharmik'/'adhyatmik' meant to our participants in the US and India, we looked for patterns of similarity and difference in their responses at the level of keywords and then clustered words that were thematically similar into categories. We followed Braun and Clarke's [68] phases of thematic analysis, including familiarization with

the responses through numerous read-throughs, identification of key words in the US and Indian data sets, categorization of the responses based on thematic similarities between the key words, and definition and naming of the categories.

Because Indians responded to the Hindi probes in both Hindi and English, we analyzed the two sets of responses separately, anticipating that the Indian English responses might be more similar to the US responses than the Indian Hindi responses. To make word searches easier, we transliterated responses in Devanagari (the Hindi alphabet) into Roman script. As we identified key terms in one language, e.g., "marga" (path) or "sanskriti" (culture) in Hindi and "denominations" and "traditions" in English, we searched for similar terms in the other language as well.

We identified and iteratively refined clusters of thematically similar terms as we continued to search for related terms in both data sets. For example, an early search clustered all the responses that included variants of 'dharma' and/or 'religion' under one thematic heading. When we later recognized that this grouping included both uncountable nouns (e.g., 'religion') and countable nouns (e.g., 'a religion' or 'religions'), we divided it to reflect this thematic distinction.

The final response categories were generated based on a shared theme, such as a type of noun or a shared reference to something (e.g., deities, belief, norms, this world/other world, self/soul/atman). If a response included two or more terms from a given category (e.g., 'God' and 'deity' or 'god' and 'higher power'), the response was only counted once. When categories were finalized, we counted the number of words in each category for each of the three groups of participants. If a response included two or more terms from different categories (e.g., God and belief) it was counted in each category. Within categories, individual search terms allow us to highlight culture-specific differences in content (e.g., in the deities named) regardless of whether the overall differences (e.g., in the number of references to deities) were significant. If the respondent did not address a question, their responses were excluded from the data.

## Transparency and openness

Following Journal Article Reporting Standards [JARS; 69] and, in keeping with JARS-Qual [70], we report on how we selected our sample pool, determined our sample size, and collected and analyzed our meta-survey data. Response data and validation templates have been made publicly available at the Open Science Framework and can be accessed at https://doi.org/10.17605/OSF.IO/W6YHG. This study's design and its analysis were not pre-registered.

## Validation results

### Experience items

We attempted to validate 60 items, most of which were drawn from an earlier, unvalidated set of items [35]. Of the 60 items tested, we successfully validated 38. Validation results for the final iteration of each of these 38 experience items are presented in Table 7. These items were understood as intended at least 85% of the time in both English (with a US population) and Hindi (with an Indian population). Complete response data, individual coder ratings, and consensus ratings for the final iteration of all 38 experience items is presented in OSF.

Items with low PPU or NPU values in one of the populations are also indicated in Table 7. Table 8 presents a summary of responses that contributed to low PPU and NPU values. Generally speaking, low PPU or NPU values were due either to a misinterpretation on the part of participants that we were unable to eliminate with revised wording of the item or to one or two incorrect responses that represented a large fraction of the total number of "Yes" responses

**Table 7. Validation outcomes for 38 experience items.**

| # | Nickname | Group | United States | | | | India | | | | Iterations (US) | Iterations (IN) | Extent of Change[a] |
|---|----------|-------|---------------|---|-----|-----|-------|---|-----|-----|------|------|------|
| | | | Proportion Understood | 3s | PPU | NPU | Proportion Understood | 3s | PPU | NPU | | | |
| 1 | Joy | Emotion | 20/20 (100%) | 1 | 100% | 100% | 22/22 (100%) | 0 | 100% | 100% | 1 | 1 | None |
| 2 | Peace | Emotion | 18/18 (100%) | 0 | 100% | 100% | 20/20 (100%) | 0 | 100% | 100% | 3 | 3 | Minor |
| 3 | Love | Emotion | 24/25 (96%) | 1 | 100% | 86% | 19/20 (95%) | 2 | 92% | 100% | 5 | 3 | Major |
| 4 | Loss | Emotion | 19/20 (95%) | 0 | 93% | 100% | 21/21 (100%) | 0 | 100% | 100% | 2 | 2 | Minor |
| 5 | Awe | Emotion | 21/21 (100%) | 0 | 100% | 100% | 21/22 (95%) | 0 | 93% | 100% | 1 | 1 | None |
| 6 | Fear | Emotion | 20/20 (100%) | 0 | 100% | 100% | 18/19 (95%) | 1 | 91% | 100% | 1 | 1 | None |
| 7 | Hopelessness | Emotion | 20/20 (100%) | 0 | 100% | 100% | 18/20 (90%) | 0 | 88% | 100% | 2 | 2 | Minor |
| 8 | Misfortune | Emotion | 21/21 (100%) | 0 | 100% | 100% | 19/20 (95%) | 1 | 91% | 100% | 1 | 1 | None |
| 9 | Compassion | Emotion | 19/20 (95%) | 0 | 93% | 100% | 21/21 (100%) | 0 | 100% | 100% | 5 | 4 | Major |
| 10 | Pleasure | Emotion | 21/21 (100%) | 0 | 100% | 100% | 21/22 (95%) | 1 | 93% | 100% | 1 | 3 | Minor |
| 11 | Places (special) | Emotion | 19/19 (100%) | 1 | 100% | 100% | 20/21 (95%) | 1 | 94% | 100% | 2 | 2 | Minor |
| 12 | Devotion (object) | Emotion | 18/18 (100%) | 2 | 100% | 100% | 18/18 (100%) | 1 | 100% | 100% | 2 | 4 | Minor |
| 13 | Devotion (people) | Emotion | 20/20 (100%) | 2 | 100% | 100% | 22/22 (100%) | 0 | 100% | 100% | 3 | 1 | Minor |
| 14 | Light(s) | Sensory/Body | 20/21 (95%) | 0 | 100% | 94% | 18/20 (90%) | 0 | 75% | 93% | 4 | 2 | Minor |
| 15 | Sounds (Voices) | Sensory/Body | 19/20 (95%) | 0 | 80% | 100% | 20/20 (100%) | 0 | 100% | 100% | 3 | 3 | Major |
| 16 | Touch | Sensory/Body | 23/23 (100%) | 2 | 100% | 100% | 20/22 (91%) | 1 | 71% | 100% | 3 | 2 | Minor |
| 17 | Faces | Sensory/Body | 19/20 (95%) | 0 | 100% | 83% | 17/20 (85%) | 0 | 100% | 57% | 2 | 3 | Minor |
| 18 | Paralysis | Sensory/Body | 18/19 (95%) | 0 | 100% | 93% | 21/21 (100%) | 0 | 100% | 100% | 2 | 1 | Minor |
| 19 | Pain | Sensory/Body | 21/22 (95%) | 0 | 94% | 100% | 15/17 (88%) | 1 | 86% | 82% | 3 | 3 | Major |
| 20 | Absorbed | Sense of Self | 20/20 (100%) | 0 | 100% | 100% | 20/22 (91%) | 1 | 94% | 75% | 1 | 1 | None |
| 21 | OBE | Sense of Self | 18/19 (95%) | 0 | 100% | 92% | 17/18 (94%) | 2 | 83% | 100% | 1 | 1 | None |
| 22 | Diminished Self | Sense of Self | 20/20 (100%) | 0 | 100% | 100% | 20/20 (100%) | 0 | 100% | 100% | 3 | 4 | Minor |
| 23 | Automaticity | Sense of Self | 18/20 (90%) | 0 | 100% | 88% | 17/19 (89%) | 2 | 71% | 100% | 3 | 3 | Minor |
| 24 | Presence (nonordinary) | Presence | 20/20 (100%) | 0 | 100% | 100% | 21/22 (95%) | 0 | 80% | 100% | 1 | 2 | Minor |
| 25 | Guidance | Presence | 16/18 (89%) | 2 | 80% | 92% | 18/18 (100%) | 3 | 100% | 100% | 3 | 2 | Major |
| 26 | Places (animated) | Presence | 17/19 (89%) | 0 | 100% | 85% | 20/20 (100%) | 1 | 100% | 100% | 1 | 3 | Minor |
| 27 | Objects (animated) | Presence | 20/20 (100%) | 0 | 100% | 100% | 22/22 (100%) | 1 | 100% | 100% | 2 | 1 | Minor |
| 28 | Lucid Dreaming | Abilities | 20/20 (100%) | 1 | 100% | 100% | 20/21 (95%) | 5 | 86% | 100% | 2 | 2 | Minor |
| 29 | Deja vu | Abilities | 18/21 (86%) | 2 | 83% | 89% | 19/22 (86%) | 2 | 72% | 93% | 3 | 3 | Minor |
| 30 | Past Life | Abilities | 19/20 (95%) | 1 | 100% | 94% | 19/19 (100%) | 3 | 100% | 100% | 3 | 2 | Minor |
| 31 | ESP (events) | Abilities | 19/22 (86%) | 1 | 83% | 88% | 21/23 (91%) | 0 | 75% | 95% | 4 | 3 | Major |
| 32 | ESP (minds) | Abilities | 19/19 (100%) | 1 | 100% | 100% | 22/22 (100%) | 0 | 100% | 100% | 6 | 5 | Major |
| 33 | Healing | Sickness/Health | 28/31 (90%) | 1 | 86% | 92% | 20/22 (91%) | 2 | 100% | 100% | 6 | 5 | Major |
| 34 | Near Death | Sickness/Health | 20/21 (95%) | 2 | 100% | 91% | 22/22 (100%) | 1 | 100% | 100% | 1 | 2 | Minor |
| 35 | Coincidences | Meaning | 17/19 (89%) | 1 | 100% | 71% | 20/21 (95%) | 1 | 100% | 93% | 1 | 3 | Minor |
| 36 | Messages | Meaning | 19/20 (95%) | 2 | 88% | 100% | 21/21 (100%) | 0 | 100% | 100% | 1 | 1 | None |
| 37 | Deep Insight | Meaning | 16/17 (94%) | 1 | 100% | 89% | 19/20 (95%) | 1 | 88% | 100% | 2 | 3 | Minor |

*(Continued)*

**Table 7.** (Continued)

| # | Nickname | Group | United States | | | | India | | | | Iterations (US) | Iterations (IN) | Extent of Change[a] |
|---|---|---|---|---|---|---|---|---|---|---|---|---|---|
| | | | Proportion Understood | 3s | PPU | NPU | Proportion Understood | 3s | PPU | NPU | | | |
| 38 | Meaning in Life | Meaning | 21/21 (100%) | 0 | 100% | 100% | 22/25 (88%) | 1 | 94% | 75% | 1 | 1 | None |

[a]A Minor change involves added emphasis or clarification without a substantive change in the intended interpretation of the item. A Major change involves a substantial change in the wording of the item as a result of broadening, narrowing, or otherwise changing the intended interpretation of the item.

to rarely experienced items. Here we provide a brief narrative summary of the validation results.

**Emotion items.** We were able to validate four items (*Joy*, *Fear*, *Awe*, and *Misfortune*) with no revisions in either English or Hindi, two items (*Pleasure* and *Devotion [people]*) with no changes in one language and minor changes in the other, and five items (*Peace*, *Loss*, *Hopelessness*, *Places [special]*, and *Devotion [object]*) with minor changes, e.g., adding emphasis, in both languages. The *Love* and *Compassion* items, which were initially combined, required major changes in both languages. *Compassion*, which went through 5 iterations in English and 4 in Hindi, was the most challenging due to respondents' tendency to conflate compassion as a trait (e.g., "I am a compassionate person") with an experience of compassion that stood out from other such experiences. The final iteration of the item replaces "had an experience" with "can recall a specific experience", "compassion" with "compassion for the suffering of others (human or nonhuman)," and underscores "stood out from other such experiences." This wording elicited single instances of the sort we intended from all but one respondent, who said s/he felt "soul crushing" experiences of compassion "every time I see people begging on the street for food." We marked this response as "understood" because it was clear that the person had repeated experiences of compassion that stood out (i.e., felt "soul-crushing") in a particular context ("every time I see people begging on the street for food") from experiences of compassion in other contexts.

**Sensory / Body items.** We were able to validate four items (*Light[s]*, *Touch*, *Faces*, and *Paralysis*) with only minor changes in both languages. We changed the ending on "I have perceived light or lights" from "for which there seemed to be no obvious cause" to "no ordinary physical source," after testing to see how those who believe in God and/or the paranormal would characterize them as causes. With other sensory items (*Sounds [voices]* and *Touch*), we were able to substitute "when it did not seem like anyone was really there" for "no obvious cause." *Sounds (voices)* required more major changes, which involved focusing the item more specifically, i.e., hearing voices rather than "noises, voices, or music." The PPU for *Sounds (voices)* in the US was 80%, due to one positive ('Yes') response that described a thought rather than a voice (Batch 3, P9). The PPU value for *Touch* was 71% in India, where 2/7 'yes' responses gave examples in which the source of the touch was clearly identified but in one case the touch (by the breeze) did not have the qualities of a physical human touch and in the other the person was touched by a person who was physically present. The NPU value for *Faces* was 57% in India where 3/7 who responded 'no' either said they did not understand the item or offered a poor paraphrase and were unable to provide an example. The *Pain* item underwent a major change in intended interpretation, from a broad understanding of pain to a focus on physical pain, due to idiosyncrasies in participants' interpretations of "pain" (iteration 1) and "pain (mental or physical)" (iteration 2).

**Sense of self items.** One item (*Absorbed*) required no changes and three (*OBE*, *Diminished Self*, and *Automaticity*) required minor changes. Because *Absorbed*, which asked if people

**Table 8. Items with PPU or NPU below 80% and reasons for caution.**

| # | Nickname | Group | United States PPU | United States NPU | India PPU | India NPU | Responses Rated "Not Understood" |
|---|---|---|---|---|---|---|---|
| 14 | Light(s) | Sensory/ Body | 100% (4/ 4) | 94% (16/ 17) | **75%** (3/ 4) | 93% (15/ 16) | In India, two participants described metaphorical light, rather than the experience of perceiving a light or lights. One "No"-responder described a realization (Batch 1, P1), and one "Yes"-responder described a "glow on our face when we are very happy" (Batch 4, P1).<br>In the US, one "No" described a vision of the Virgin Mary that did not specifically involve light(s) (Batch 2, P2). |
| 16 | Touch | Sensory/ Body | 100% (5/ 5) | 100% (18/18) | **71%** (5/ 7) | 100% (15/15) | In India, one "Yes" described "a cool breeze that was touching as if trying to talk to me," rather than a touch that felt like it was coming from another person (Batch 3, P10). Another "Yes" described being touched by a human who was physically present (Batch 4, P4). |
| 17 | Faces | Sensory/ Body | 100% (14/14) | 83% (5/ 6) | 100% (13/13) | **57%** (4/ 7) | In India, three participants who said "No" stated that they did not understand the question (Batch 2, P4; Batch 3, P4 & P6).<br>In the US, one participant who said "No" paraphrased the item as "a man-made face" and failed to give a specific example (Batch 1, P1). |
| 20 | Absorbed | Sense of Self | 100% (19/19) | 100% (1/ 1) | 94% (17/ 18) | **75%** (3/ 4) | In India, two respondents (one "Yes"; Batch 3, P2; and one "No"; Batch 3, P4) talked about not wasting time while completing tasks, but they did not mention any phenomenological features of becoming absorbed in the task. |
| 23 | Automaticity | Sense of Self | 100% (4/ 4) | 88% (14/ 16) | **71%** (5/ 7) | 100% (12/12) | In India, two "Yes"s described sleep paralysis rather than automatic actions (Batch 3, P5; Batch 4, P6).<br>In the US, one "No" did not count their experience of sleepwalking (Batch 1, P5). Another "No" described an out-of-body experience in their paraphrase and example (Batch 2, P6). |
| 29 | Deja vu | Abilities | 83% (10/ 12) | 89% (8/ 9) | **71%** (5/ 7) | 93% (14/ 15) | In India, one "Yes" described seeing something for the first time (Batch 1, P2). Another "Yes" did not give an example of deja vu (Batch 1, P4). One "No" described an unpleasant event that one cannot forget.<br>In the US, one "Yes" described going to a familiar place and "knowing how things work" (Batch 4, P3). Another "Yes" said "deja vu" in the paraphrase, but did not describe a specific experience (Batch 4, P6). One "No" described feeling comfortable in a strange situation (Batch 2, P5). |
| 31 | ESP (events) | Abilities | 83% (5/ 6) | 88% (14/ 16) | **75%** (3/ 4) | 95% (18/ 19) | In India, one "Yes" did not give a clear example (Batch 2, P4). One "No" described a premonition, rather than perception at a distance (Batch 1, P5).<br>In the US, two "No"s described making logical inferences about nearby events (e.g., things being closed for the holidays; Batch 3, P2; Batch 4, P3). One "Yes" described a premonition, rather than perception at a distance (Batch 4, P5). |
| 35 | Coincidences | Meaning | 100% (12/12) | **71%** (5/ 7) | 100% (6/ 6) | 93% (14/ 15) | In the US, two "No"s described single events rather than coincidences, e.g. "a death, marriage, or having a child" (Batch 3, P4) and "If you meet someone cool" (Batch 3, P7).<br>In India, one "No" stated that they did not understand the item (Batch 3, P2). |
| 38 | Meaning in Life | Meaning | 100% (10/10) | 100% (11/11) | 94% (16/ 17) | **75%** (6/ 8) | In India, one "No" described an experience of lacking meaning in life (Batch 4, P1). Another "No" only described complete "Enlightenment", using the example of Buddha's enlightenment (Batch 3, P10). One "Yes" failed to describe a single, sudden, discrete experience (Batch 4, P3). |

Batch and participant numbers (e.g., Batch 1, P1) refer to entries in the Response Data—Experience Items spreadsheet on OSF. Examples of Hindi responses are translated into English.

have had "an experience in which [they were] completely absorbed in what [they were] doing and unaware of the passage of time," is a common experience, we rated responses that referred to multiple experiences in the context of a specific activity as understood. The NPU for *Absorbed* was 75% in India because one of the four negative responses included a poor paraphrase and example. The PPU for *Automaticity* was 71% in India because 2/7 positive responses confused automaticity with sleep paralysis.

**Presence items.** Three Presence items (*Presence [nonordinary]*, *Places [animated]* and *Objects [animated]*) required minor changes in at least one language; *Guidance* required major

changes. *Guidance* had a final PPU of 80% in the US. One out of five positive responses in the US was rated as Not Understood because they described an experience that itself seemed nonordinary, rather than an experience in which they felt they were guided by a nonordinary power. *Presence (nonordinary)* had a PPU of 80% in India; one out of five positive responses in India was rated as Not Understood due to a poor paraphrase ("to ignite the power resting within") and example (experienced whenever the respondent "watch[ed] a super-hero movie."). *Presence (dead)* was understood by at least 85% of participants in both the US and India, but two out of the four "Yes"-responses in India referred to a deity or spirit rather than a dead person, resulting in a PPU of 50% in India. Thus, despite the high overall proportion of respondents who understood the item as intended, this item was removed due to the risk of skewed cross-cultural comparisons, where the prevalence in India is likely to be inflated relative to the prevalence in the US.

*'Nonordinary' in Presence items.* All of the final Presence items in English included a generic reference to "nonordinary" forces, powers, presences, and/or beings, which we hoped would encompass and uniquely designate the wide range of supernatural entities, psychic powers, and spiritual forces that were of interest to us. This turned out to be the case. In the three instances where a respondent failed to understand an item in its final iteration, they did not mention any sort of nonordinary force, power, or presence; in the case of Guidance, the one response rated as Not Understood in English (mentioned above) simply referred to a nonordinary experience, and in the case of *Places (animated)*, the 2/13 responses rated as Not Understood in English simply referred to nonordinary places. Not only was the distinction between ordinary and nonordinary forces, powers, and beings understood as we intended, but the paraphrases demonstrated that "nonordinary" captured the diversity of entities, powers, and forces of interest to us. Thus, for example, the paraphrases of "nonordinary forces or beings" for the *Presences (nonordinary)* item included a supernatural being or presence (5), God (4), ghosts (4), spirits (3), angels, demons, aliens, an unexplained nonhuman entity, an invisible being, an imaginary being, beings beyond the veil, as well as paranormal [forces] and vibrations on other levels.

Two of the Presence items in Hindi–*Presences (nonordinary)* and *Places (animated)*– included the word *asadharan*, which is usually translated as extra- or non-ordinary. The other two items that included "nonordinary" in English–*Guidance* and *Objects (animated)* were translated less literally. (A back translation of *Objects (animated)* reads: "I have come in contact with "an image (pratima), idol (murti), or other physical object *that appears to be alive.*" A back translation of Guidance reads: "I felt as if some other force was directing or guiding me.") There was only one instance in which a respondent failed to understand any of the four items in its final iteration. The respondent paraphrased *Presences (nonordinary)* as "To ignite the power resting within" and indicated that they experience this whenever they watch a "super-gero movie" [sic] (phrase inserted in English). We rated this response as a 4 (Probably not understood) because the paraphrase is poor, and, although it is possible to imagine that this respondent experiences a non-ordinary presence every time they watch a superhero movie, a more plausible interpretation is that they interpreted the item more broadly than we intended, so to include any depiction of a non-ordinary being, rather than the felt presence. The translated paraphrases of the Presences (nonordinary) item included invisible (6), supernatural (2), and divine (1) powers or energies; gods, spirits, or ghosts (4); beyond our understanding, impossible (2), unusual (2) uncontrolled by humans or machines, also paranormal, otherworldly.

**Abilities and/or paranormal powers items.** Three items (*Lucid Dreaming*, *Deja vu*, and Past Life) required only minor changes in both languages and two (ESP [minds] and ESP [events]) required major changes. Participants failed to understand our original ESP items,

which were worded fairly generically, as intended. After many iterations, we tested two more specifically worded items drawn from Irwin [24], which we were able to validate. The PPU of *Deja Vu* in India was 72%. Two of the seven positive responses were rated as Not Understood due to poor examples. The PPU of *ESP (Events)* in India was 75%, due to one out of four positive responses who offered an unclear paraphrase and example.

**Sickness and health items.** The *Near Death* item required no change in the US and only minor change in India and was well understood in both populations. The *Healing* item was inspired by healings attributed to nonmedical interventions, such as prayer or the manipulation of energies; it required major revisions and repeated iterations to arrive at wording that reliably captures the sudden and unexpected experiences that we anticipate are the mostly likely to be attributed to nonmedical interventions.

**Meaning items.** Two items in this group (*Messages* and *Meaning in Life*) required no changes and the other two (*Coincidences* and *Deep Insight*) required only minor changes. Overall, these relatively common experiences were well understood. The NPU of *Coincidences* in the US was 71.4%. Two of the seven negative responses did not involve two things happening at the same time, an important aspect of the intended interpretation of coincidence. The NPU of *Meaning in Life* in India was 75%. Two of the eight negative responses were rated as 'Not Understood'. One paraphrased the item as "enlightenment" and gave the Buddha's experience as an example; the other apparently read it backwards, as an experience of losing all meaning in life.

Twenty experience items were not understood as intended at least 85% of the time in English, Hindi, or both. For the wording of all the unvalidated items and further details on each of the items that we failed to validate, see S2 Appendix.

## Follow-up items

Validation results for 6 of the final follow-up items (*Mental State*, *Impact*, *Life Effect*, *Science*, *Reason*, and *Agent*), which were evaluated with a specific intended interpretation, are presented in Table 9. Respondents understood the follow up items as intended at least 85% of the time in both English (with a US population) and Hindi (with an Indian population). Complete response data for the final iteration of all validated follow-up items is presented in OSF. Results for the Category (R/S) item are discussed below.

**Mental state.** Because we were aware that some experiences occur in contexts such as meditation, worship, or prayer in which certain experiences are anticipated, we attempted to validate a response option that would capture these kinds of practices. We tested "when you had this experience, were you engaging in practices related to the experience?" When this formulation elicited a wide range of responses, including "while driving" and "expecting things to turn out a certain way," we substituted "were you doing something intended to bring about such an experience?" Some respondents who selected this option were intending to bring about *some* experience, but the experience they described was not an intended outcome of their action, e.g., in the context of looking for a job, they had an experience of Meaninglessness. In light of the sometimes-plausible responses that nonetheless ranged far beyond what we intended, we removed the practice-related response option from the Mental State follow-up item.

**Impact & life effect.** The final wording–"Overall, how much of an impact has this experience had on your life?"–captures the extent to which the experience had an effect on the person's life mentally or physically. The final wording of the Effects item–Overall, has the **lasting** effect of this experience, on your life or beliefs, been more positive or negative?–captures the valence of the long-term effect of the experience. In some cases, the responses to these two

**Table 9. Validation results for final follow-up items.**

| # | Nickname | Query & Response Options | Proportion Understood (US) | Proportion Understood (IN) | 3s (US) | 3s (IN) | 3*s (US) | 3*s (IN) |
|---|---|---|---|---|---|---|---|---|
| 1 | **Mental State** | When you had the experience, were you . . . (Select the most important) | **139/139 (100%)** | **138/140 (99%)** | **0** | **0** | **0** | **0** |
| | 1A | Using drugs or alcohol | 13/13 | 4/4 | 0 | 0 | 0 | 0 |
| | 1B | Affected by mental or physical illness | 9/9 | 7/7 | 0 | 0 | 0 | 0 |
| | 1C | Falling asleep, waking up, or exhausted | 6/6 | 11/12 | 0 | 0 | 0 | 0 |
| | 1D | Asleep (dreaming) | 9/9 | 12/12 | 0 | 0 | 0 | 0 |
| | 1E | None of the above | 83/83 | 104/105 | 0 | 0 | 0 | 0 |
| 2 | **Impact** | Overall, how much of an impact has this experience had on your life? | **42/43 (98%)** | **47/49 (96%)** | **0** | **1** | **1** | **1** |
| | 1A | Little or no impact | 16/16 | 16/16 | 0 | 0 | 0 | 0 |
| | 1B | Some impact | 10/10 | 23/25 | 0 | 1 | 1 | 1 |
| | 1C | Major impact | 16/17 | 29/29 | 0 | 0 | 0 | 2 |
| 3 | **Life Effect** | Overall, has the lasting effect of this experience, on your life or beliefs, been more positive or negative? | **144/147 (98%)** | **133/145 (92%)** | **1** | **10** | **1** | **3** |
| | 3A | Very positive effect | 48/50 | 56/58 | 1 | 5 | 1 | 1 |
| | 3B | Somewhat positive effect | 30/30 | 21/28 | 0 | 4 | 0 | 1 |
| | 3C | Neutral or no effect | 51/51 | 30/31 | 0 | 1 | 0 | 1 |
| | 3D | Somewhat negative effect | 6/7 | 17/19 | 0 | 0 | 0 | 0 |
| | 3E | Very negative effect | 9/9 | 9/9 | 0 | 0 | 0 | 0 |
| 5 | **Science** | Do you think science can explain how this experience happened? | **85/87 (98%)** | **60/60 (100%)** | **4** | **2** | **0** | **2** |
| | 5A | Yes, science can or will be able to explain it. | 51/51 | 19/19 | 0 | 0 | 0 | 1 |
| | 5B | No, something More is involved. | 34/36 | 41/41 | 4 | 2 | 0 | 1 |
| 6 | **Reason** | Why do you think it happened to you? (Select the closest answer.) | **121/121 (100%)** | **78/78 (100%)** | **1** | **0** | **0** | **6** |
| | 6A | To offer me a sign or message | 27/27 | 25/25 | 0 | 0 | 0 | 1 |
| | 6B | To reward or punish me for my actions | 10/10 | 9/9 | 0 | 0 | 0 | 2 |
| | 6C | Due to destiny/fate | 16/16 | 21/21 | 0 | 0 | 0 | 0 |
| | 6D | None of the above (may include chance/probability) | 68/68 | 23/23 | 1 | 0 | 0 | 3 |
| 7 | **Agent** | Who, if anyone, caused you to experience this? (Select the most important.) | **83/84 (99%)** | **101/102 (99%)** | **0** | **0** | **0** | **2** |
| | 7A | God or gods | 13/13 | 46/46 | 0 | 0 | 0 | 0 |
| | 7B | Other spiritual beings or forces (including the dead) | 14/14 | 9/10 | 0 | 0 | 0 | 0 |
| | 7C | None of the above | 56/57 | 46/46 | 0 | 0 | 0 | 2 |

Follow-up item 4, Cause (R/S), was not evaluated with respect to an intended interpretation.

queries overlap; in others, however, they diverge. For example, in response to the Despair item, one respondent said 'yes' because "when my wife left me, it was the most hopeless I had ever been in my entire life." He indicated that the experience had "very much impact," because "it turned my world upside down." The effect was only "somewhat negative," because "there have been happy times since it has happened, life hasn't totally stunk." It should be noted that respondents may choose "neutral or no effect" for experiences that had either "little or no impact" or "major impact," if the lasting effect was equally positive and negative.

**Science.** This item was well understood as originally worded. The intended interpretation required respondents to make a distinction between "science" and something "More," without us stipulating definitions of either. Most respondents made some kind of distinction between "science" and something "More," but they did not have–and we did not require–a common understanding of "science", the "More", or the nature of the boundary between them. Many

who said 'yes' alluded to specific scientific explanations that fit their experience rather than referring to science in general without specifying where they would draw the boundary. In some cases where respondents explicitly specified the boundary, they placed it differently, e.g., some included personal feelings and intuitions in the More and others included them in what psychology could explain. Although the reference to "More" hearkens back to William James and was interpreted by most as a reference to spiritual or supernatural powers, we accepted any interpretation of "More" that went beyond science as the respondent understood it.

**Reason.**   Apart from specifying that "None of the above" included chance and probability, this item was well understood as originally worded.

**Agent.**   This item underwent extensive revision. We initially conceived this item as identifying a wide range of religious or spiritual causes to which they might attribute their experience. Causes initially included God, gods, or other deities; extraordinary or paranormal powers or abilities (including your own); other extraordinary or supernatural beings; cosmic spiritual forces, processes, or principles; and normal mental or physical processes, including chance. Respondents had difficulty distinguishing many of these response options from each other and from natural processes. For instance, "Cosmic spiritual forces, processes, or principles" was sometimes interpreted as referring to phenomena at the edges of physical science, such as quantum physics and Einstein's "spooky action at a distance". In light of these responses, we focused the item on two types of Agents, both of which validated easily: "God or gods" and "other spiritual beings or forces (including the dead)."

**Category (R/S).**   The goal of validation for the Category (R/S) item was to understand how participants in the US and India interpreted the terms "religious," "dharmik," "spiritual," and "adhyatmik," rather than to demonstrate that the item is understood as intended across both populations. Therefore, we did not evaluate the proportion of participants who understood each item as intended. Rather, we investigated the ways in which each participant interpreted the item. We had 50 US responses to the questions: "What do you think 'religious' means?" and "What do you think 'spiritual' means?" We had 64 Indian responses to the questions: "'Dharmik'–is shabda se aap kya samajhte hai? Apne shabdo mei batae." (What do you think that 'dharmik' means?) and "'Adyhatmik'–is shabda se aap kya samajhte hai? Apne shabdo mei batae." (What do you think that 'adhyatmik' means?) Of the 64 Indian responses, 50 answered in Hindi and 14 in English. In reporting the results, we refer to the latter as Indian Hindi responses and Indian English responses respectively. For the data, see Response Data–Category (R/S) in OSF. In translating the Hindi responses in the data file, we left key terms (i.e., dharma, bhagwan [God], atman [self/soul]) untranslated.

*'Religious' and 'Dharmik'.* Of the 14 Indian English responses to the Hindi probe, 5 indicated that 'dharmik' meant 'religious.' One Indian Hindi respondent also translated 'dharmik' as 'religious' by inserting the English word in their response. In addition to these direct translations, we identified two major types of responses in the data: responses that defined 'religious' or 'dharmik' in terms of religion(s) or dharma(s), which we could further classify as either countable (concrete) or uncountable (abstract) nouns, and responses that connected 'religious' or 'dharmik' with something (i.e., deities, beliefs, beliefs plus, morality/norms, or practices/devotions) (see Table 10).

Countable Nouns refer to identifiable entities in the world, e.g., specific beliefs, practices, or social groups; Uncountable Nouns refer to abstract concepts that may have many levels of meaning and do not refer to specific beliefs, practices, or groups. We classified a noun as Countable if it was plural (e.g., religions) or was preceded by an article (a, ek, kisi), an indefinite pronoun (e.g., all, any, whichever), or a possessive adjective (my, your, one's, apane, unake, hamari). Abstract nouns lacking these features, i.e., 'religion' and 'dharma', are classified as Uncountable. The Nonordinary Beings category includes specific deities (e.g., God,

**Table 10. Interpretations of "religious" and "dharmik" in the US and India and proportion of respondents in each category.**

| Category | Search Terms | US | $IN_{Total}$ | $IN_{Hindi}$ | $IN_{Eng}$ | $\chi^2$ |
|---|---|---|---|---|---|---|
| | | | Proportion of Respondents in Category | | | |
| Uncountable Noun | religion<br>dharma, dharam, dharm | 3/51 | **17/64** | 15/50 | 2/14 | 8.45** |
| Countable Noun | [a] religion(s), [a] denomination, [a] tradition, [a] faith<br>[ek, kisi, hamari/hum, unake, apane] dharm/a | **15/51** | 8/64 | 8/50 | 0/14 | 5.07* |
| Religious or Dharmik connected to: | | | | | | |
| Nonordinary Beings | God/god(s), higher power, deity, divine, creator; bhagwan, devata/devatao, ishwar ke icca | **23/51** | 17/64 | 14/50 | 3/14 | 4.66* |
| Belief | belief, believing, believe<br>manyata, manta, maanthe, maanana, maanee, manne/a, wala yakeen, yakin rahta | **22/51** | 11/64 | 8/50 | 3/14 | 9.34** |
| Belief Plus | follow/ing/er, (having) faith; ka chalte hain, shraddha, biswas, vishwas, aastha, samarpit ho, dhyan rakhata/dhyan rakhta and khush rakhana [when applied to deities/dharma] | 12/51 | 14/64 | 12/50 | 2/14 | 0.044 |
| Morality/Norms | morals, norm, values; niyam (norm), mooly/moolyon (values), acchai/bhalai (good deeds/goodness); sahi karma, kaam (right action/deed), jathi (caste), raaste/marg (path), acha/bura (good/bad), sanskrit (culture), jeevan tarika (way of life) | 3/51 | **14/64** | 13/50 | 1/14 | 5.76* |
| Practices | practices, church, prayer; bhakti, puja | **9/51** | 3/64 | 3/50 | 0/14 | 5.10* |

Search terms are given for each category. Proportion of responses in each category is given for all US responses, all Indian responses ($IN_{Total}$), Indians responding in Hindi ($IN_{Hindi}$), and Indians responding in English ($IN_{English}$). Chi square values compare the proportion of US responses with the proportion of $IN_{Total}$ responses in each category. Where there is a significant difference in proportions between the US and India$_{Total}$, the higher proportion is in bold.

*Significant at p < 0.05

** Significant at Bonferroni-corrected p < 0.0071

bhagwan), generic terms referring to deities (e.g., gods, devata), terms used to characterize deities (e.g., creator, divine) and other nonordinary entities (e.g., paranormal beings), as well as more expansive concepts that are not necessarily personified (e.g., higher power). We distinguished between "belief" in the sense of an affirmation, e.g., "I believe God exists" (the Belief category), and terms, such as "following (a deity)" or "biswas/vishvas" (having faith or trust [in a deity])" that imply belief but add an active relational dimension that goes beyond a simple affirmation (the Belief Plus category). As one American explained, "religious means following a religion by not only believing in it, but also following it." The Morality/Norms category includes direct references to morals and norms, value-laden terms (e.g., good, bad, right), and norm-infused concepts (e.g., marga [path] jeevan tarika [way of life], sanskriti [culture]). Practices includes generic references to practices, as well as specific practices, e.g., prayer, puja.

A series of Chi-Square tests were used to determine whether the proportion of responses in each category differed between US and Indian respondents. To correct for multiple comparisons, Bonferroni-corrected alpha levels 0.05/7 = 0.0071 were used. See Table 10 for raw counts of category membership for each group, including the subsets of Indian respondents who responded in Hindi (IN-H) and those who responded in English (IN-E). US participants were significantly more likely than Indian participants to define 'religious'/'dharmik' in terms of belief. Indian participations were significantly more likely than US participants to define 'religious'/'dharmik' in relation to an Uncountable Noun ('religion'/'dharma').

*'Spiritual' and 'Adhyatmik'.* Of the 14 Indians who responded in English, 4 said 'adhyatmik' meant 'spiritual,' two without further elaboration. One Indian who responded in Hindi inserted 'spiritual' in English. Three Indian Hindi respondents said (the adjective) 'adhyatmik' was connected to (the noun) 'adhyatma' or to 'spirituality' inserted in English, but none who responded in English in the US or India did so. Most Americans and most Indians (whether responding in Hindi or English) defined 'spiritual' or 'adhyatmik' in terms of a connection to

**Table 11. Interpretations of "spiritual" and "adhyatmik" in the US and India and proportion of respondents in each category.**

| Category | Search Terms | US | IN$_{Total}$ | IN$_{Hindi}$ | IN$_{Eng}$ | $\chi^2$ |
|---|---|---|---|---|---|---|
| | | \multicolumn Proportion of Respondents in Category | | | | |
| Something Inward | soul, atman, aatm, inner/internal (andhar/ andar), essence | 13/51 | 14/64 | 10/50 | 4/14 | 0.21 |
| This World | nature, earth, world (duniya, aayat), universe, people, prithvi (earth), sabhi insaano (all humans) | 7/51 | 5/64 | 4/50 | 1/14 | 1.06 |
| Other World | realm, dimension, other world / otherworld, world (excluding 'this world'), aloukik/alaukik (other world) | **10/51** | 2/64 | 2/50 | 0/14 | 8.25** |
| Nonordinary Beings | God/god(s), divinity, higher being(s), higher or supreme power, bhagwan, devata (gods), prabhu (lord), spiritual entities, paranormal beings | 14/51 | 16/64 | 14/50 | 2/14 | 0.088 |
| Force/Energy | shakti, force, energy, power (excluding higher power) | 3/51 | 8/64 | 7/50 | 1/14 | 1.43 |
| Knowledge | knowledge, ideas, maan ki shakti, gnan, gyan, bodh | 1/51 | 5/64 | 5/50 | 0/14 | 1.97 |

The search terms are given for each category. The proportion of respondents that characterized 'spiritual' and 'adhyatmik' using the search terms is given for US responses, all Indian responses (IN$_{Total}$), Indians responding in Hindi (IN$_{Hindi}$), and Indians responding in English (IN$_{English}$). Chi square values compare the proportion of US responses with the proportion of IN$_{Total}$ responses in each category. Where there is a significant difference in proportions between the US and IN$_{Total}$, the higher proportion is in bold.

*Significant at $p < 0.05$

** Significant at Bonferroni-corrected $p < 0.0071$

something outward and/or inward (see Table 11). The Something Inward category included generic references to something inner or internal (andhar/andar) and specific internal concept, i.e., "soul" and "atman/aatm." The This World category included references to the natural world, e.g., nature, earth (prithvi), universe, and its inhabitants, i.e., people, sabhi insaano (all humans). The Other World category included references to other world(s) (aloukik/alaukik) or realms. The Nonordinary Beings category includes specific deities (e.g., God, bhagwan), generic terms referring to deities (e.g., gods, devata), terms used to characterize deities (e.g., creator, divine) and other nonordinary entities (e.g., paranormal beings), as well as more expansive concepts that are not necessarily personified (e.g., higher power, divinity). The Force/Energy category includes references to force, energy, or power (excluding higher power) and shakti (force/energy, excluding the goddess Shakti). The Knowledge category includes references to knowledge (gnan, gyan, bodh) and ideas.

A series of Chi-Square tests were used to determine whether the proportion of responses in each category differed between US and Indian respondents. To correct for multiple comparisons, Bonferroni-corrected alpha levels $0.05/7 = 0.0071$ were used. All p-values were greater than 0.05 except for the Other World category. US participants were more likely to define 'spiritual' in relation to 'Other World' (10/51) than were Indian participants (2/64), $\chi^2(1) = 8.25$, $p = 0.004$, and this difference remained significant using the Bonferroni-corrected p-value of 0.0071.

## Discussion

Researchers have struggled to compare experiences across cultures. Doing so is particularly challenging given the way that culture infuses people's lived experience and in many cases the constructs that researchers use to study them. The Inventory of Nonordinary Experiences (INOE) is designed to avoid discipline-specific theoretical constructs (i.e., imposed etics). The feature-based approach on which the INOE is premised [13] separates experience items that query phenomenological features of lived experiences from follow-up items that query

additional aspects of the lived experience. The INOE shifts the basis of comparison to generically-worded items (proposed etics) and tests these items to ensure that they are understood as intended in the cultures to be compared. If they are, then these items can serve as a stable basis for comparison between those cultures (derived etics).

The two-stage design of the INOE is premised on two assumptions that we tested here. First, subjects must be able to recognize the generically-worded features (proposed etic) in their own lived experiences (emic). Second, when subjects say "Yes" to a generically-worded experience item, they must have a specific lived experience in mind that allows them to answer follow-up questions regarding the context, effects, and appraisals of the experience. In other words, the design is premised on our ability to strip experiences, described phenomenologically, of culture-laden concepts, while at the same time ensuring that items remain comprehensible. To test these two assumptions of the INOE we used the RPE method, a novel technique for item-level validation [14, 15].

In the absence of established expectations for sample size for cognitive interviews or web probing, we developed an iterative process for revising items until they are understood as intended (or eliminated from the survey), created a transparent framework for reporting overall evaluations of responses, and identified items where there is increased risk of either false negatives or false positives. The transparent reporting framework allows researchers to replicate the process and confidently "borrow" items from surveys. Most critically, use of the RPE method to validate survey items ensures that survey results measure actual differences in a construct rather than differences in how the items are understood. In doing so, it adds to the power of psychological survey research.

## Deriving etics: Experience items understood as intended

Using the RPE method, we found that English-speaking participants in the US and Hindi-speaking participants in India reliably recognized a generically-worded feature in 38 of the 60 experience items tested. Further, when a participant endorsed an item, they had a specific lived experience in mind or, in a few cases, a specific *type* of experience (see "Cautions" section below). Thus, these items can function as derived etics for comparing nonordinary experiences in these two cultures. More broadly, this shows that it is possible to validate generically-worded items using the RPE method; this constitutes a proof of concept and offers a new methodological approach to cross-cultural comparison in the context of large-scale survey research.

**Emotion items.** In light of constructionist approaches to emotions, such as Barrett's Conceptual Act Theory [71], one of the project's reviewers raised the possibility that the items in our emotion group would prove too cross-culturally variable for us to validate. This did not prove to be the case. That the Emotion items we tested were consistently understood by a large proportion of participants offers evidence that the emotion words reflect similar feelings (i.e., subjective emotional experiences) in both the US and India, and can be understood as intended across these two linguistic groups based on that common feature.

**Presence items.** In formulating some of these items, we were able to refer to the kind of presences we intended with specific wording. This was the case with *Guidance* and *Objects (animated)* in Hindi. In other cases, we needed a more general way to refer to the wide range of supernatural entities, psychic powers, and spiritual forces that were of interest to us. To do that, we referred to "non-ordinary" forces, powers, presences, and/or beings in four of the English Presence items and "asadharan" in two of the Hindi Presence items. The validation process allowed us to test whether this formulation was understood as intended. The paraphrases of the *Presence (nonordinary)* and *Places (animated)* items and the wide range of appropriate examples indicated that the intended meaning of "non-ordinary" and "asadharan"

was well understood when used to modify forces, powers, entities, and/or presences. This suggests that in some cases "nonordinary" or its Hindi equivalent "asadharan" may be a useful starting point for generically identifying nonphysical entities and forces that are commonly described using culture-specific terms (e.g., God, spirits, or ghosts). When these terms are used in the item, it indicates that subjects viewed the presences as "nonordinary"; if other terms are used researchers can't assume that people consider the presences as "nonordinary."

**Sense of Self items.** The analysis of the four Sense of Self items that we were able to validate (discussed here) and the nine we were not (discussed in the next section) suggests that some types of changes in sense of self may be more amenable to cross-cultural survey study than others. In discussing our ability to validate the Sense of Self items, we can distinguish between items that refer to the disruption of high-level reflective processing (the narrative self) and those that refer to lower-level processes (e.g., "minimal" or "embodied" self; for reviews, see [72–74]). The former includes those items that involve a decrease in self-related thought, the loss of access to semantic autobiographical information, and discontinuities in the sense of personal identity. The latter includes changes in self-location, body ownership, or body awareness (for a discussion, see [74, 75]).

Of the ten items that involved disruption of the narrative self, we were only able to validate *Absorbed*, which involves a decrease in the frequency of self-related thought. Yaden et al. [76] consider absorption as the most common (routine, ordinary) form of self-transcendent experience, which they define as "transient mental states marked by decreased self-salience and increased feelings of connectedness ["with other people or one's surroundings"]." *Absorbed* had two features that allowed people to identify the experience: a focus on an activity ("I was completely absorbed in what I was doing"), and the post-hoc recognition of their lack of awareness of the passage of time ("and unaware of the passage of time").

We were able to validate the three items that did not refer to disruptions in the narrative self. *OBE* involves a change in self-location (". . . it seemed as if I left my physical body"); *Diminished Self* involves a change in bodily awareness ("I have felt small or insignificant in relation to something vast or powerful"), and *Automaticity* involves a change in body ownership (". . . it seemed like my body was performing actions outside my control. . ."). In so far as the narrative self was unaffected by the experience, this likely made the experience easier to observe and report than items involving a disruption in the narrative self. Experiences that involve a disruption in the narrative self may thus pose unique challenges for measurement and cross-cultural comparison, as discussed below.

**Cautions.** Our evidence for the validity of some items is stronger than others. Although a substantial portion of items (29/38) were understood as intended by 85% or more of participants, a minority of items (9/38) had a low PPU or NPU score in one of the two populations (see Table 8), including two of the items just discussed: *Absorbed* and *Automaticity*. In these cases, we have evidence that the experiences are relatively well-recognized in the tested population, which supports the claim that the features queried exist in both populations, and thus can be considered derived etics. However, in practice these items may pose difficulties for cross-cultural comparison. It is likely that for some of these items, certain false positive or false negative responses may be more common in one population than another, creating a systematic bias in cross-cultural comparisons. Researchers should consider whether items with a high rate of false positive or false negative responses would pose a problem for their intended use of the INOE, especially if it is being used to compare US with Indian populations.

In the case of *Absorbed* and occasionally with a few other items, people responded "Yes" to an item based on a recurring experience, such as "I'm a dancer so it happens a lot. During practice when I'm in the groove." In these cases, we did not require participants to report a specific, singular experience. Rather, we allowed for responses that referred to experiences that

repeatedly occurred in a particular context. It is possible that this may lead to biases in responses to the follow-up items (e.g., responding to different follow-up items based on different instances, or considering the overall effect of multiple experiences rather than a single experience). Responses to follow-up items should be interpreted with this in mind, and future research should investigate the potential effect of this lack of specificity on responses to follow-up items.

## Reasons why some items failed to validate

Twenty-two items were removed from the INOE because we lacked confidence that they were understood as intended in both the US and India. There are several possible reasons why we may have been unable to validate an item. (1) It may mean that we asked about a feature that is not present or too rare to be reliably recognized in the population of interest. (2) It may mean that the item failed to describe a feature in a way that could be recognized by that population. The item may have been worded too broadly, such that it elicited a number of features rather than a common feature, or worded so that it elicited metaphorical interpretations rather than phenomenological features. (3) It may be that the feature is present and the item is understood as intended, but our probes failed to elicit sufficient evidence of validity from the respondents.

If a feature is present in a population, it may yet be possible to construct and validate an item querying the feature in that population. But it is often hard to tell why we were unable to validate an item, and, thus, whether an item we failed to validate could be validated, or refined and validated, with further effort. As a result, we can't necessarily make conclusive statements about whether a feature is absent, or not widely recognizable, in a certain population, based on our inability to collect sufficient evidence of validity.

In some cases, however, our evidence suggests that certain widely used concepts, such as mystical experiences, spirit possession, and kundalini experiences, do not have consistent phenomenological features that can offer a suitable basis for cross-cultural comparison; and that questionnaires purporting to measure such experiences using multi-item measures may be capturing a much wider range of experiences than intended. Of the nine Sense of Self items that we were unable to validate, the five that involve "ego dissolution" were adapted from questionnaires designed to measure mystical experiences, and the four that involve "self-alien" intrusions into the self are commonly associated with possession-type experiences. Our Energy item, which we were also unable to validate, attempted to capture the core feature associated with kundalini experiences.

**Mystical experience-related items.**    There is a long history of theological and philosophical reflection on experiences characterized as "mystical" with little consensus on what is meant by the term (for an overview, see [77]). Historians in the Catholic tradition, which is the source of the term, have used it to refer to experiences of the presence of God [78]. Other scholars [e.g., 79, 80] define the concept broadly, such that it is largely synonymous with "religious experience." William James's [81] well-known definition narrowed the focus to experiences that are ineffable, noetic, transient, and passive (i.e., involuntary), but, as he indicated, could include both drug-induced and pathological experiences. Philosophers and theologians progressively narrowed James's definition over the course of the 20th century, eliminating the drug-induced and psychopathological variants to arrive at an even narrower definition [for discussion, see 82]. This narrowed definition typically characterizes a mystical experience as a highly positive experience in which the sense of self disappears or is absorbed into something larger; philosophers refer to this as an experience of "undifferentiated unity" [83], "the pure consciousness experience" [84], or "absolute unitary beings" [85]. The two most commonly used mysticism scales–Hood's Mysticism Scale [86] and the Mystical Experience

Questionnaire [MEQ; 87, 88]–both operationalize Stace's experience of "undifferentiated unity." Researchers, aware of the phenomenological overlap between ostensibly positive experiences of undifferentiated unity and negatively valenced experiences of "ego dissolution," incorporated items from the M-scale and MEQ in the recently developed Ego Dissolution Inventory [89].

Four of the five items we drew from these scales attempted to capture this sort of disruption in the narrative self (*Unity*, *Connectedness*, *All Disappears*, and *NonExistence*). Both *Unity* and *Connectedness*, which we tested in various and sometimes overlapping formulations, involved a dissolution of the sense of self and loss of individual identity. *All Disappears* ("everything seemed to disappear from my mind until nothing remained") and *NonExistence* ("I have felt as if I no longer existed") not only involve a reduction in self-related thought, but also the inability to access semantic autobiographical information. In the course of validation, we divided *Connectedness* into *Connectedness (all)* and *Connectedness (others)* to see if distinguishing between a more cosmic sense of becoming "part of a greater whole that extends far beyond me" and a more social sense of "becoming one with everyone at a large group event, would elicit more consistent responses. Over the course of numerous iterations (4 US/7 IN for Unity, 4 US/6 IN for *Connectedness [All]*, and 4 US/4 IN for *Connectedness [Others]*), these items elicited a variety of responses that failed to converge on a specific common feature. The other two items–*All Disappears* and *NonExistence*–elicited mostly incomprehension and attempts to guess what was meant.

Our inability to validate these items suggests that the scales from which these items were drawn capture a much wider range of experiences than they intend and may, thus, be an ineffective way to measure the shift in the sense of self that, according to some Western traditions, is the hallmark of a mystical experience. It may be that there is a common phenomenological feature that can be characterized as an experience of "undifferentiated unity" or "ego dissolution" that could be captured by improved survey items. It is possible that the experience is too subtle to characterize in the everyday language of the lay population and that a better formulation would require the use of more specialized vocabularies better suited to describing such experiences. Or it may be that the experience of "undifferentiated unity" does not refer to a specific feature. Carhart-Harris, who was involved in creating the Ego Dissolution Inventory [89], has cautioned researchers that "'self-loss' and related expressions such as 'ego dissolution' are notoriously ambiguous notions" and recommended the multidimensional approach to specifying disruptions in sense of self adopted here [75]. Lindström et al. [74] share this concern and demonstrate the value of in-depth interviews of subjects who report experiences that involve a profound sense of self loss in teasing apart the different aspects of such experiences.

Regardless of whether it is possible to validate a more refined "Unity" or "Ego Dissolution" item for use in surveys of the general population, it should be noted that the mysticism construct, as defined by Stace and operationalized in the mysticism scales, "presupposes, on theological grounds, that experiences of undifferentiated unity are inherently positive, that positive valence is part of the experience itself, and that the experience is culturally unmediated (i.e., that the cultural environment does not play a significant role in constituting the positively valenced experience as it unfolds)" [82]. The design of the INOE deliberately separates experience items from follow up questions regarding valence and meaning so that researchers can investigate the relationship between them across cultures.

**Possession-type experiences.** We tested four items that attempted to capture self-alien experiences, that is, experiences in which aspects of the self—thoughts, feelings, or agency—do not appear to be one's own. We attempted to capture self-alien thoughts and feelings–referred to as "intrusions" or "alien control of thought" in the psychiatric literature [90–92] with the *Intuitions*, *Moods*, and *Inner Dialogues* items. Our inability to arrive at wording that separated

thoughts or feelings that seemed markedly alien from mundane conflicted thoughts or mixed emotions likely reflects the fuzziness of the concept and the variable meanings–both literal and metaphoric–that people associate with that which seems like it is "not me."

In the case of *Another Self in Body*, which attempted to capture the most marked disruptions in the narrative self, the item was well understood by those who did not endorse it, many of whom gave possession as an example of such an experience. However, it was not well understood by those who did endorse it. Of the few (6/45) who claimed to have had such an experience, most described a co-conscious, conscience-like "inner voice" and/or a sense of inner conflict or being "of two minds." Moreover, the INOE as a self-report measure can only capture experiences in which subjects are co-conscious with (rather than displaced by) the alien self [13]. This suggests that, while the cultural concept of another (alien) self in the body is widely understood in the general population, the subjective experience of the few who claim to have had the experience is not consistent. The work of ethnographers and historians, however, would suggest that even in a larger sample of people whose behaviors are locally characterized as "spirit possession" [93–95] or "avesta" (Sanskrit/Hindi: [96, 97]), we would likely have found a variety of subjective sensations, which they or those around them appraise as self-alien. Moreover the sense of switching between "selves" can be present in various contexts where it is understood in different ways, e.g., when bicultural individuals switch cultural frames [98] or actors perform in theaters [99].

**Kundalini experience.**   We worded the *Energy* item ("I have experienced flows of energy within my body, for which there seemed to be no ordinary explanation") in an attempt to express the sensation that ostensibly characterizes the "kundalini experience" in generic terms. The failure of respondents to understand "flows of energy" consistently in either context led us to conclude that the idea of a "kundalini experience" is a cultural concept that–like possession–is not (and perhaps cannot be) reliably connected with specific subjective sensations at least in the general population. It may be that advanced practitioners cultivate particular sensations that could be formulated with more specialized vocabulary. The concept of *shakti*, which is translated as energy/power, is deeply embedded in Hindu systems of thought, but is not necessarily associated with a consistent subjective experience. Depending on the context, it can refer to energy in a secular sense, the energy that radiates up through the chakras in a kundalini experience, or to the goddess Shakti. This suggests that there are many possible "energy-like" experiences, some of which in some contexts are characterized as "kundalini experiences." Qualitative studies of energy-like somatic experiences, which have been reported across cultures, also suggest that the underlying phenomena are heterogeneous [100].

## Takeaways regarding "religious"/"dharmik" and "spiritual"/"adhyatmik" in the US and India

Researchers often translate surveys with potentially culture-specific concepts, such as "religious" or "spiritual," into other languages and administer them, assuming that the meanings of the terms and their translations are relatively stable and generalizable across cultures. By collecting qualitative data from both populations of interest in the validation process, we investigated the extent to which the standard Hindi translations of *religious* and *spiritual* were actually equivalent to the English concepts. Our investigation revealed a complex pattern of similarities and differences from which we can glean the following key points.

**The American and Indian participants' interpretations of *religious* and *dharmik* differed in significant ways.**   With respect to *religious* and *dharmik*, we found that American respondents were more likely than Indian respondents to think of "religious" in terms of "religions" (traditions or denominations or faiths), that is, concrete social groups and specific

practices, such as attending church and prayer. Americans were also more likely to define "religious" in terms of belief than were the Indians. This is in keeping with the Protestant cultural bias of US samples, in which traditional groups form around specific religious beliefs, and highlights the importance of investigating the meaning of culture-laden terms in the contexts in which they are used. In contrast, we found that Indians were more likely than American participants to think of "dharmik" in terms of the abstract noun "dharma," which can be translated not only as "religion," but also as righteousness, duty, justice, law, or ethics, and to associate dharma with morality and norms [40]. Although *religious* and *dharmik* are often treated as equivalents in comparative studies, the difference in the American and Indian understandings is congruent with the different range of meanings traditionally associated with "religion" and "dharma" that specialists in South Asian religions have long noted [40, 41].

**In contrast to this, American and Indian participants interpreted "spiritual" and "adhyatmik" in similar ways.** In both the US and India, respondents tended to view spiritual/adhyatmik as related either to something inward (i.e., spirit/soul in the US and *atman* in India) or to a deity (a generic higher power in the US and *bhagwan* in India). Neither term was linked to institutions or groups. These basic findings are congruent with large scale comparisons of the meaning of spirituality/spiritualität in the US and Germany [101]. Although there were no significant differences between the US and India in the number of responses that fell within five of our six categories, US participants were significantly more likely than Indians to associate spiritual/adhyatmik with another world or realm. This difference is not due to the absence of other worlds–of which there are thousands–in Hinduism and other South Asian traditions. It is, however, consistent with a difference in traditional spiritual goals that may make "other worlds" more salient for Americans than Indians when defining spiritual/adhyatmik. Thus, the traditional Christian goal is salvation, i.e., eternal life in heaven (a postulated place); the traditional Hindu goal is enlightenment (*moksha*), in which the *atman* is freed from the cycle of birth and rebirth (*samsara*) ([102], s.v. moksha). Although the cycle of birth and rebirth may include a temporary stay in one of many other realms, these realms are mere waystations not the goal of the spiritual path and the ultimate goal (*moksha*) is not necessarily associated with a place or realm ([103], s.v. heaven). This difference, along with the cultural specificity of terms, such as *soul* and *God*, on the one hand, and *atman*, *bhagwan*, and *shakti*, on the other, should alert researchers to underlying differences in Christian and Hindu cosmologies, despite the similarity in participants' understandings of "spiritual" and "adhyatmik."

**The wide range of meanings associated with "religious" and "spiritual" in the US and "dharmik" and "adhyatmik" in India make it difficult to offer a single definition of either term in either population.** In this regard, researchers should note that, although some participants explicitly linked each of the four terms with deities, many others did not. Thus, when participants characterize an experience as R/S or D/A, researchers cannot assume that they believe that deities were involved. The additional follow up items allow researchers to assess the extent to which responses to the R/S and D/A item correlate with scientific explanation, teleological reasoning, and the presence of supernatural agents.

## Limitations and future directions

Because both the design of the INOE and the use of the RPE method are new, we will discuss limitations and future directions, first, with respect to the INOE as a tool for studying nonordinary experiences and, second, with respect to the RPE method as a tool for the item level validation of surveys.

**The INOE as a flexible tool for studying nonordinary experiences.** We believe the INOE with its feature-based design holds much promise as a tool for researchers interested in

studying nonordinary experiences both within and across cultures. Here we have offered evidence that the items in this published version of the INOE are understood as intended in a general English-speaking population in the US and a general Hindi-speaking population in India. We used MTurk workers as a proxy for the general population in each country, recognizing that neither sample is completely representative of the population of their country. For example, MTurk users in the United States have been shown to skew male, as in the current study, as well as to score higher on measures of negative affect and social anxiety [104, 105]. MTurk also has well-known issues with data quality [63, 106], even relative to other online survey platforms [107]. In light of these limitations, we recommend that researchers consider alternative online survey platforms, such as Prolific, which may yield higher-quality data.

Additionally, MTurk workers in India might be more Westernized than the general Indian population, potentially minimizing observed differences. However, any differences we observe in spite of this potential bias constitute strong evidence of a true difference between US and Indian populations. Given these constraints, researchers can use the validated INOE to compare experiences based on phenomenological features rather than discipline-specific theoretical constructs in the US and India. Because each item in the INOE is individually validated, researchers do not need to use the INOE as a whole, but can select the items that are most relevant, given their research agenda. Researchers can use it to make comparisons between the US and India, to compare subpopulations within either country, or to compare subpopulations between the two countries.

For those researchers interested in using the validated INOE, either to study populations in the US or India or to compare populations in the US and India, we want to stress that our relatively small sample of 20 participants limits researchers' ability to observe rare misinterpretations or cross-cultural differences in interpretation, which may be present in substantial subsets of responses in large-scale survey studies. We urge particular caution in deciding whether to use the 10 items for which we failed to reach both a PPU and an NPU of 80% in either the US or India for cross-cultural comparisons. A summary of the responses rated Not Understood for validated items is presented in Table 8. Items with a PPU or NPU value below 80% in either the US or India are indicated there and with an asterisk in the item list in S1 Appendix.

Researchers may also expand on our efforts in important ways. First, researchers can see if items we were unable to validate could be validated with further efforts. In some cases, it may just be a matter of finding better wording. In other cases, the experience, however well worded, may not be consistently understood by a lay population. Researchers may want to test such items with specialized subpopulations, such as psychedelic drug users (who may be more familiar with the experience being described), or Buddhist meditators (who may share a vocabulary for describing such experiences). Researchers could also focus on a set of specialized subpopulations that report similar experiences to identify the phenomenological features associated with them. For example, they could focus on those who report specific alterations in sense of self to identify specific sensations that they tend to appraise as self-alien.

Second, researchers can construct and validate new experience items or follow-up items. For particular studies, researchers may want to add newly developed experience items to a selection of experience items drawn from the INOE. Alternatively, they may want to develop new follow-up questions to add to or substitute for the ones we developed. The last four follow-up items reflect our interest in what counts for people as a religious or spiritual experience in light of the many different ways these concepts are understood and the variety of experiences to which they are applied. For those who share that interest, we would note that the religion-related follow-ups were tested in two large complex cultures and are likely skewed toward such cultures; it may be difficult to translate and validate these items in small scale

cultures. Other researchers may be interested in exploring other cultural categorization schemes, such as what counts as "political" or what counts as "evidence."

Finally, we encourage researchers to use the RPE to validate the INOE in additional languages and cultures around the globe. In doing so, researchers may not want to translate and validate all the items; they may want to focus on particular items and/or add some new ones. To enable comparisons between cultures, however, we encourage researchers to translate and validate items in light of our intended interpretations. If this proves difficult, e.g., if an item has to be substantially revised in order to be understood in the new culture, the revised item would need to be re-validated in the US and/or India before responses could be compared.

**The RPE method as a tool for validating surveys.** The value of the RPE method extends beyond this particular application. Existing surveys commonly do not report evidence of validity based on the response process. The omission of this type of evidence makes it difficult to tell if a survey measures what the researchers intend to measure. As a result, researchers may be unable to tell whether variation in responses is due to actual variation in what is being measured, or whether it is due to differences in how the items are understood. The RPE method provides a transparent and replicable means of reducing this uncertainty. We encourage researchers to apply the RPE methods to the surveys they use or create, whether they are designed for use across cultures or within a single population prior to assessing other aspects of validity, such as the factor structure, reliability coefficients, correlations with existing scales, etc. Templates for organizing responses and coding data are available in OSF.

In our case, the RPE method allowed us to scrutinize participants' interpretations of items in more depth than traditional quantitative methods of validation allow and eliminate many misinterpretations of items. Our inability to validate some items drawn from other surveys, such as Mysticism Scale [86] and the Mystical Experience Questionnaire [87], suggests that they may be capturing a much wider range of experiences than researchers intended. Wider use of the RPE method would allow researchers to refine their items and ensure that their survey results are due to the differences that they intend to measure. In applying the RPE method, we would encourage researchers to consider collecting a larger number of participants. A relatively small sample of 20 participants for the final iteration of each item limits researchers' ability to observe rare misinterpretations or cross-cultural differences in interpretation, which could be present in substantial subsets of responses in large-scale survey studies (especially if members of these subpopulations were not well-represented in from the validation sample). A larger sample would also produce more precise estimates of the proportion of participants who understood each item, both overall and among those who responded "Yes" (PPU) or "No" (NPU).

As this concluding section indicates, this article can be read in two ways: as introducing a new tool for comparing culturally-laden lived experiences between populations and as a case study that illustrates the value of a new tool for item level validation of surveys. As a tool for comparing experiences between populations, the separation of phenomenological features from follow-up appraisals offers a new means of investigating the potentially universal (etic) and culture-specific (emic) aspects of lived experiences that does not rely on discipline-specific theoretical constructs. A bottom-up approach to item validation allows participants to aid in the refinement of the items (proposed etics) upon which researchers hope to base their comparisons. This approach ensures that the proposed common feature is understood as intended in both populations and can serve as a derived etic. Because it does not seek to measure an overall construct, the INOE is unusually flexible. Researchers can draw from it, add to it in light of their research agendas, and feel more confident borrowing items for use in their own research. As a case study, the article illustrates the potential value of the RPE method for creating surveys in which the items are understood as intended and, thus, to increase the likelihood that survey results are due to the differences that researchers intend to measure.

## Supporting information

**S1 Appendix. Validated items and intended interpretations.**
(PDF)

**S2 Appendix. Unvalidated items.**
(PDF)

## Acknowledgments

The authors are grateful to Nikita Vyas for her technical support and to our many research assistants—Bhavi Bhagat, Gurleen Basra, Laura Deutsch, Ojas Dewan, Victoria Galvan, Eli Kirkeeng, Ankita Lakhotia, Mazi Lala, Anvi Mittal, Sahaj Parikh, Emily Pollock, Danielle Sanchez, and Savannah Tellander–for their dedicated work on the project.

## Additional materials in OSF

https://osf.io/w6yhg/?view_only=c845f3e182eb43539817220fdf63498b
Templates
Response Data—Experience Items
Response Data—Follow-up Items
Response Data—Category (R/S).

## Author Contributions

**Conceptualization:** Ann Taves, Elliott Ihm, Michael Barlev, Michael Kinsella.

**Data curation:** Elliott Ihm, Melissa Wolf.

**Formal analysis:** Ann Taves, Elliott Ihm, Melissa Wolf.

**Funding acquisition:** Ann Taves, Elliott Ihm, Melissa Wolf, Michael Barlev.

**Investigation:** Elliott Ihm, Melissa Wolf, Maharshi Vyas.

**Methodology:** Ann Taves, Elliott Ihm, Melissa Wolf.

**Project administration:** Ann Taves, Elliott Ihm.

**Software:** Elliott Ihm, Melissa Wolf.

**Supervision:** Ann Taves, Elliott Ihm, Maharshi Vyas.

**Validation:** Ann Taves, Elliott Ihm, Melissa Wolf, Maharshi Vyas.

**Visualization:** Ann Taves, Elliott Ihm.

**Writing – original draft:** Ann Taves, Elliott Ihm, Melissa Wolf.

**Writing – review & editing:** Ann Taves, Elliott Ihm, Melissa Wolf, Michael Barlev.

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
