## [Decision Letter · Decision Letter 0]

31 May 2023

PONE-D-23-05816

The Inventory of Nonordinary Experiences (INOE): Evidence of validity in the United States and India

PLOS ONE

Dear Dr. Taves,

Thank you for submitting your manuscript to PLOS ONE. After careful consideration, we feel that it has merit but does not fully meet PLOS ONE’s publication criteria as it currently stands (MINOR REVISIONS). Therefore, we invite you to submit a revised version of the manuscript that addresses the points raised during the review process.

We look forward to receiving your revised manuscript.

Kind regards,

Luca Valera

Academic Editor

PLOS ONE

Journal Requirements:

**Additional Editor Comments:**

Dear dr. Taves,

I am happy to inform you that your paper has been accepted with minor revisions. I am sorry for the long delay, but we had some problems with the search of reviewers, due to the topic of your paper. I hope you can revise your the reviewer's commentaries and suggestions and send the paper back to the journal. Best regards,

Luca Valera

Reviewers' comments:

Reviewer's Responses to Questions

**Comments to the Author**

1. Is the manuscript technically sound, and do the data support the conclusions?

Reviewer #1: Yes

Reviewer #2: Yes

2. Has the statistical analysis been performed appropriately and rigorously? 

Reviewer #1: Yes

Reviewer #2: Yes

3. Have the authors made all data underlying the findings in their manuscript fully available?

Reviewer #1: Yes

Reviewer #2: Yes

4. Is the manuscript presented in an intelligible fashion and written in standard English?

Reviewer #1: Yes

Reviewer #2: Yes

5. Review Comments to the Author

Reviewer #1: The manuscript deals with a new approach to the intercultural study of non-ordinary experiences (NOE). Especially, the INOE provides a first inventory of NOE, gathering US and Hindi cultures and checking their etic and emic aspects. I think that the topic is of paramount importance, in order to exceed the biases stemming from a priori established constructs and the ethnocentric perspective of scientific studies, as well as the traditional positivist approach to comparative philosophy and religion.

The manuscript is rather complex, but it is well-written and I have read it with interest. I think that the adopted method and aims are relevant and innovative, and should be pursued in future studies. I would only provide some suggestions that, in my opinion, may improve the quality of discussion. Anyway, the authors are not required to follow them, should they consider them irrelevant:

• The problem of non-ordinary experiences has been introduced by Ludwig in 1966 with the name Altered State of Consciousness, gathering both physiological and pathological conditions (Ludwig, 1966; Vaitl et al., 2005). Later on, the Concept of Anomalous Experiences as well as the umbrella term Exceptional Human Experience have been introduced (Cardeña et al., 2014; Palmer & Hastings, 2013). Their heterogeneity, lack of coherence and comprehensiveness of these classifications has recently led to revise them and introduce the term Non-Ordinary Mental Expressions (NOMEs), including a new synthetic classification and emphasizing the links between different NOMEs (Facco et al., 2021).It seems to me that a short description of the topic as a whole in the introduction might improve the quality of the paper by showing the presence and overlap between different non-ordinary experiences within a given culture as well as their relevance in cross-cultural comparisons.

• The authors’ approach seems to me perfectly in line with the metaphilosophical approach, aimed to search for key concepts and meanings common to different philosophies, beyond their formal differences and different modes of theorization (Burik, 2009; Jullien, 2015, 2016; Overgaard et al., 2013; Pasqualotto, 2008; Weber, 2013). Such an approach implies the openness toward the “other” and the willingness to step outside one’s own comfort zone. I have previously reported on the need for a transcultural, metaphilosophical approach to both consciousness and Self (Facco et al., 2023; Facco, Al Khafaji, et al., 2019), and even to foreign politics (Facco & Tagliagambe, 2022), an inescapable fact in a globalized world. Actually, the terms Ego, I, Self and soul are ill defined and their meaning greatly overlaps in the history of philosophy, while several scientific studies skip relevant aspect of the Self that may be encompassed and better understood by a more openminded approach including the knowledge of other cultures. Therefore, I have greatly appreciated the authors’ perspective aiming to check the etic-emic components, as a tool to overcome the narrow ethnocentric and chronocentric stance of most scientific studies: it seems to me a relevant approach that may be viewed as the other side of the coin of metaphilosophy.

• Absorption is not non-ordinary in itself, given that we spend our life incessantly shifting between different levels of arousal, awareness, absorption, dissociation, distraction, drowsiness, sleep and dream. Absorption is defined as the total attention that fully engages one’s representational resources, leading to imperviousness to distracting events (Tellegen & Atkinson, 1974). Therefore, absorption – i.e., concentration on a task - can be considered as the reciprocal aspect of dissociation - i.e., the neglect of what is outside the focus of attention (Facco, 2022). It is worth considering that absorption is an essential aspect of both meditation and hypnosis. Absorption also involves an openness to absorbing and self-altering experiences and, thus, the capacity to engender many NOMEs, including mystical experiences, OBEs, Near-death like experiences and experiences of other identities (with their link with dissociative identity disorders and possession) (Facco, Casiglia, et al., 2019; Facco, Mendozzi, et al., 2019; Machado et al., 2022; Shaw et al., 2023). An altered sense of agency (including automaticity) is a feature of hypnosis as well (Haggard & Clark, 2003). I think that reframing the comments on absorption (p. 55) taking into account the above comments, might improve the quality of discussion.

In short, what is perceived in different ways in different cultures belongs to universal mind faculties and experiences, both the East and the West; some of them look non-ordinary from the adopted worldview, a fact regarding their interpretation rather than the experiences in themselves. Here, the etic-emic and the metaphilosophical approaches seem to me two inseparable aspects of the same, wiser way to comprehend both human mind and culture-related differences. Both of them allow to overcome cultural filters and constraints stemming from any narrow ethnocentric and chronocentric perspective(the latter involved when studying ancient thought).

I have found a few minor flaws:

1. Table 1, “ANNEX”: perhaps AANEX?

2. P. 16, l. 6, Mental state: It seems to me that it could be implemented with the item“Other”; in fact, these kind of experiences may be also favored by a variety of factors non mentioned in the text, such as psychotropic drugs.

3. TP. 29, l. 10, the formula and the following similar formulas are not clear to me ( I only see squares), and I am afraid it will not be clear to the reader as well:

4. P. 44, l. 1, “mediation”: perhaps “meditation”?

References

Burik, S. (2009). Opening philosophy to the world: derrida and education in philosophy. Educational Theory, 59(3), 297–312. https://doi.org/10.1111/j.1741-5446.2009.00320.x

Cardeña, E., Lynn J., S., & Krippner, S. (Eds.). (2014). Varieties of Anomalous Experiences. (2nd ed.). American Psychological Association.

Facco, E. (2022). Hypnosis and Hypnotic ability between old beliefs and new evidences: an epistemological reflection. American Journal of Clinical Hypnosis, 64(1), 20–35. https://doi.org/10.1080/00029157.2020.1863181

Facco, E., Al Khafaji, B. E., & Tressoldi, P. (2019). In search of the true self. Journal of Theoretical and Philosophical Psychology, 39(3), 157–180. http://dx.doi.org/10.1037/teo0000112

Facco, E., Casiglia, E., Al Khafaji, B. E., Finatti, F., Duma, G. M., Mento, G., Pederzoli, L., & Tressoldi, P. (2019). The neurophenomenology of out-of-body experiences induced by hypnotic suggestions. International Journal of Clinical and Experimental Hypnosis, 67(1), 39–68. https://doi.org/10.1080/00207144.2019.1553762

Facco, E., Fracas, F., & Tagliagambe, S. (2023). Consciousness and the mind-brain-body-world relationship: towards a transdisciplinary and transcultural approach. Advances in Social Sciences Research Journal, 10(1), 414–433. https://doi.org/10.14738/ASSRJ.101.13896

Facco, E., Fracas, F., & Tressoldi, P. (2021). Moving beyond the concept of altered state of consciousness: the Non-Ordinary Mental Expressions (NOMEs). Advances in Social Sciences Research Journal, 8(3), 615–631. https://doi.org/10.14738/ASSRJ.83.9935

Facco, E., Mendozzi, L., Bona, A., Motta, A., Garegnani, M., Costantini, I., Dipasquale, O., Cecconi, P., Menotti, R., Coscioli, E., & Lipari, S. (2019). Dissociative identity as a continuum from healthy mind to psychiatric disorders: epistemological and neurophenomenological implications approached through hypnosis. Medical Hypotheses, 130(109274), 1–11. https://doi.org/10.1016/j.mehy.2019.109274

Facco, E., & Tagliagambe, S. (2022). The Century-Old Swinging of Russia Between Order and Dissolution, East and West: A Historical and Psychocultural Insight in the Russian-Ukrainian War. Advances in Social Sciences Research Journal, 9(11), 447–460. https://doi.org/10.14738/ASSRJ.911.13497

Haggard, P., & Clark, S. (2003). Intentional action: conscious experience and neural prediction. Conscious.Cogn, 12(1053-8100 (Print)), 695–707.

Jullien, F. (2015). De l’Etre au Vivre. Lexique eurochinoise de la pensée. Gallimard.

Jullien, F. (2016). Il n’y a pas d’Identité Culturelle mai Nous Défendons les Resources d’une Culture. Éditions de L’Herne.

Ludwig, A. M. (1966). Altered states of consciousness. Arch.Gen.Psychiatry, 15(0003-990X (Print)), 225–234.

Machado, S. A. F., Farinha, A. P., & Simões, M. (2022). Hypnotically Induced Near-Death-Like Experiences: An Exploratory Study of Phenomenological Similarities to Near-Death Experiences. Journal of Near-Death Studies, 40(1), 47–68. https://doi.org/10.17514/JNDS-2022-40-1-p47-68

Overgaard, S., Gilbert, P., & Burwood, S. (2013). Introduction: what good is metaphilosophy? In An Introduction to Metaphilosophy. (pp. 1–10). Cambridge University Press. https://doi.org/https://doi.org/10.1017/CBO9781139018043

Palmer, G., & Hastings, A. (2013). Exploring the Nature of Exceptional Human Experiences: Recognizing, Understanding, and Appreciating EHEs. In The Wiley‐Blackwell Handbook of Transpersonal Psychology (pp. 331–351). https://doi.org/doi:10.1002/9781118591277.ch18

Pasqualotto, G. (2008). East & West. Marsilio Editori.

Shaw, J., Gandy, S., & Stunbrys, T. (2023). Transformative effects of spontaneous out of body experiences in healthy individuals: An interpretative phenomenological analysis. Psychology of Consciousness: Theory, Research, and Practice., in press. https://doi.org/https://doi.org/10.1037/cns0000324

Tellegen, A., & Atkinson, G. (1974). Openness to absorbing and self-altering experiences (“absorption”), a trait related to hypnotic susceptibility. J.Abnorm.Psychol., 83(0021-843X (Print)), 268–277.

Vaitl, D., Birbaumer, N., Gruzelier, J., Jamieson, G. A., Kotchoubey, B., Kubler, A., Lehmann, D., Miltner, W. H., Ott, U., Putz, P., Sammer, G., Strauch, I., Strehl, U., Wackermann, J., & Weiss, T. (2005). Psychobiology of altered states of consciousness. The Psychological Bulletin, 131(0033–2909), 98–127.

Weber, R. (2013). “How to Compare?” – On the Methodological State of Comparative Philosophy. Philosophy Compass, 8(7), 593–603.

Reviewer #2: This paper is an important contribution to the methodology of cross-cultural studies of unusual/nonordinary/anomalous experiences. The RPW methodology provides a systematic way of assessing the cross-cultural validity and comparability of these experiences. The INOE will likely be used, at least partially, in future studies by other scholars.

My recommended changes are, thus, of a minor nature:

1) Although the small N is mentioned as a limitation, there are additional limitations that should be mentioned: a) most respondents are male, despite the fact that typically females endorse more most of these experiences; b) although the procedure make it unlikely that respondents were robots or non-attentive, nonetheless the problems with Mechanical Turk should be mentioned, e.g., Perspectives on Psychological Science: "Too Good to Be True: Bots and Bad Data From Mechanical Turk." by Margaret A. Webb and June P. Tangney, c) it is VERY good that a question about mental states was added, but the list of possibilities is small and does not contain, for instance, meditation.

2) The literature review is fairly comprehensive. I would nonetheless recommend these two items: 1) Acta Psychiatrica Scandinavica, Evidence for the early clinical relevance of hallucinatory-delusional states in the general population, R. Nuevo, J. Van Os, C. Arango, S. Chatterji, J. L. Ayuso-Mateos, for cross-cultural differences in a type of unusual experience (hallucination), 2) Dissolution of What? The Self Lost in Self-Transcendent Experiences. Lindström, L., Kajonius, P. & Cardeña, E., 2022, In: Journal of Consciousness Studies. 29, 5-6, p. 75-101, for an example of teasing out the conceptual vagueness of the term "ego dissolution."

3) It is mentioned on page 60 that the mysticism construct, as defined by Stace and

operationalized in the mysticism scales, “presupposes, on theological grounds, that experiences

of undifferentiated unity are inherently positive, that positive valence is part of the experience

itself, and that the experience is culturally unmediated (i.e., that the cultural environment does

not play a significant role in constituting the positively valenced experience as it unfolds)” (76), but that is not the conclusion reached by Wulff (reference 71), that mentioned that such experiences are not necessarily positive and that culture PARTLY mediates the experience. This should be mentioned, as well as his (and others}) categorization of different types of mystical experiences, which may explain part of the problem with the question used.

4) Re. the item Another self in the body", given the small and technologically savy sample, the problem may be that no one was involved in ritual possessions and thus could not attest to this. And even in ritual possessions there are differences in, e.g., amount of amnesia, see Cardeña, E., Schaffler, Y., & Van Duijl, M. (2023). The other in the self: Possession, trance, and related phenomena. In M. J. Dorahy, S. N. Gold, & J. A. O’Neill (Eds.), Dissociation and the dissociative disorders: Past, present, future. 2nd ed. (pp. 421-432). Routledge. doi: 10.4324/9781003057314-31

5) The references list needs to be proofed better. For instance, capitalization of paper titles is quite inconsistent.

6. PLOS authors have the option to publish the peer review history of their article (what does this mean?). If published, this will include your full peer review and any attached files.

Reviewer #1: **Yes: **Enrico Facco

Reviewer #2: No

---

## [Author Response · Author response to Decision Letter 0]

2 Jun 2023

NOTE: The content pasted here is from the uploaded "Response to Reviewers." Pasting it here eliminated much of the formatting that made our response easy to read. 

Response to Reviewers 

We are grateful to the editor and reviews for their helpful comments. We have indicated our responses in blue text and manuscript edits in red text.

Editors Requests

a. Minor edits were made in the title page to more precisely match the style requirements. We have two corresponding authors since AT is better positioned to respond to those with concerns about framing and discussion (such as the reviewers of the article) and EI is better positioned to respond to those with concerns about the details of the validation process.

b. Changes in the body of the text included adjusting the line spacing between headings and text so it is consistently double spaced, formatted the table titles per the style requirements, and adjusted the formatting of the S1 and S2 Appendices. We also made minor clarifications in the headers and captions for Tables 10 and 11.

2. Additional details regarding participant consent were added in the Methods section and copied to the Ethics Statement in the submission form. 

3. Review of reference list. We did not spot any problems with the references, apart from some formatting issues, e.g., inconsistent capitalization, which we corrected. We unmasked citations to our own work and added the following citations:

a. Arditte, K. A., Çek, D., Shaw, A. M., & Timpano, K. R. (2016). The importance of assessing clinical phenomena in Mechanical Turk research. Psychological Assessment, 28(6), 684–691. https://doi.org/10.1037/pas0000217

b. Barlev, M., Taves, A., & Kinsella, M. (2021). Mapping nonordinary experiences across cultures in the U.S. and India. https://doi.org/10.31234/osf.io/fvc6w

c. Cardeña, E., Schaffler, Y., & van Duijl, M. (2023). The other in the self: Possession, trance, and related phenomena. In M. J. Dorahy, S. N. Gold, & J. A. O’Neil (Eds.), Dissociation and the dissociative disorders: Past, present, future (Second edition, pp. 421–432). Routledge, Taylor & Francis Group.

d. Douglas, B. D., Ewell, P. J., & Brauer, M. (2023). Data quality in online human-subjects research: Comparisons between MTurk, Prolific, CloudResearch, Qualtrics, and SONA. PLOS ONE, 18(3), e0279720. https://doi.org/10.1371/journal.pone.0279720

e. Facco, E. (2023). Consciousness and the mind-brain-body-world relationship: Towards a transdisciplinary and transcultural approach. Advances in Social Sciences Research Journal, 10(1). https://doi.org/10.14738/assrj.101.13896

f. Facco, E., Fracas, F., & Tressoldi, P. (2020). Moving beyond the concept of altered state of consciousness: The Non-Ordinary Mental Expressions (NOMEs) [Preprint]. MindRxiv. https://doi.org/10.31231/osf.io/b5wyf

g. Lindström, L., Kajonius, P., & Cardeña, E. (2022). Dissolution of What? The Self Lost in Self-transcendent Experiences. Journal of Consciousness Studies, 29(5), 75–101. https://doi.org/10.53765/20512201.29.5.075

h. Ludwig, A. M. (1966). Altered States of Consciousness. Archives of General Psychiatry, 15(3), 225. https://doi.org/10.1001/archpsyc.1966.01730150001001

i. McCredie, M. N., & Morey, L. C. (2019). Who Are the Turkers? A Characterization of MTurk Workers Using the Personality Assessment Inventory. Assessment, 26(5), 759–766. https://doi.org/10.1177/1073191118760709

j. Webb, M. A., & Tangney, J. P. (2022). Too Good to Be True: Bots and Bad Data From Mechanical Turk. Perspectives on Psychological Science, 174569162211200. https://doi.org/10.1177/17456916221120027

Reviewer #1 (Enrico Facco)

We are grateful to Enrico Facco for providing “some suggestions that… may improve the quality of discussion. … the authors are not required to follow them, should they consider them irrelevant.”

• Overview of the topic: After reviewing some of the categories under which researchers study NOEs, Facco suggests “a short description of the topic as a whole in the introduction might improve the quality of the paper by showing the presence and overlap between different non-ordinary experiences within a given culture as well as their relevance in cross-cultural comparisons.” 

Response: In introducing the paper, we mention both the cultural concepts that are built into reports of experiences and the categories lay people and researchers use to classify them. We decided to elaborate on the former (the cultural concepts built into reports of experiences) in the introduction rather than the categories researchers use to classify them, because it allowed us to illustrate the distinction between the phenomenology of the experience (they heard a voice) from the appraisal (it was God or an auditory hallucination) that is a fundamental feature of the INOE. We return to the categories that researchers use to categorize experiences or, more precisely to the constructs (discipline-specific theoretical ideas) that researchers use to designate an experiential object of study (p. 4), highlighting constructs such as religious, mystical, psychopathological, and anomalous experiences. Rather than rewrite the introduction, we are adding a reference to Facco et al. 2021 to note that he supports our claim that there is considerable overlap between experiences classified under different headings both within and across cultures.

The commonalities that define each set of experiences and distinguish them from one another are not phenomenological features, but culturally-derived disciplinary appraisals of what counts as religious, mystical, psychopathological, or anomalous (Saville-Smith, 2023). These disciplinary classifications tend to obscure the overlap between experiences with similar phenomenological features within a culture and at the same time limit cross-cultural comparisons (Facco et al., 2021).

Facco rightly points out that we don’t mention the “altered states of consciousness” literature, but rather than rewriting the introduction, we can easily add ASCs to the paragraph below. 

In an attempt to avoid both religious and pathological connotations, researchers have introduced more expansive constructs. Some, whose interests arose in response to the effects of psychedelic drug use, refer to altered states of consciousness. Others, whose interests descend from the psychical researchers and parapsychologists of the 20th century, have advanced constructs such as extraordinary, anomalous, or exceptional experiences. In doing so, they also rely on binaries such as normal versus altered, ordinary versus extraordinary, and everyday versus uncommon (Cardeña et al., 2000, p. 4; Fach et al., 2013, p. 1; Ludwig, 1966; Kohls & Walach, 2006, p. 126). 

• Metaphilosophy. Response: We appreciate Facco’s highlighting of the aims of metaphilosophy and the congruence between the transdisciplinary, metaphilosophical approach he has advocated and the approach we adopted. We think that Facco et al 2023 offers added evidence to support our opening claim that “researchers increasingly recognize that the mind and culture interact any many levels to constitute our lived experience” so will add that citation to the citation cluster. 

Researchers increasingly recognize that the mind and culture interact at many levels to constitute our lived experience (Constant et al., 2022; Facco et al., 2023; Kirmayer et al., 2020; Veissière et al., 2019), yet we know relatively little about the extent to which culture shapes the way people appraise their experiences and the likelihood that a given experience will be reported. 

• Absorption. Absorption is not non-ordinary in itself, given that we spend our life incessantly shifting between different levels of arousal, awareness, absorption, dissociation, distraction, drowsiness, sleep and dream. Absorption is defined as the total attention that fully engages one’s representational resources, leading to imperviousness to distracting events (Tellegen & Atkinson, 1974). Therefore, absorption – i.e., concentration on a task - can be considered as the reciprocal aspect of dissociation - i.e., the neglect of what is outside the focus of attention (Facco, 2022). It is worth considering that absorption is an essential aspect of both meditation and hypnosis. Absorption also involves an openness to absorbing and self-altering experiences and, thus, the capacity to engender many NOMEs, including mystical experiences, OBEs, Near-death like experiences and experiences of other identities (with their link with dissociative identity disorders and possession) (Facco, Casiglia, et al., 2019; Facco, Mendozzi, et al., 2019; Machado et al., 2022; Shaw et al., 2023). An altered sense of agency (including automaticity) is a feature of hypnosis as well (Haggard & Clark, 2003). I think that reframing the comments on absorption (p. 55) taking into account the above comments, might improve the quality of discussion.

In short, what is perceived in different ways in different cultures belongs to universal mind faculties and experiences, both the East and the West; some of them look non-ordinary from the adopted worldview, a fact regarding their interpretation rather than the experiences in themselves. 

Response: After reflecting on this suggested reframing of our absorption item, we decided against doing so. While there may be a correlation between absorption and dissociation in some contexts and both may be a component of mediation and hypnosis, we found this too speculative for our purposes. In the discussion, we wanted to stick as closely as possible to the phenomenology of the experiences. We included Absorbed in the set of items that involve an alteration in the sense of self. This highlights an aspect of absorption (i.e., a decrease in self-related thought) that is not highlighted in Telegen & Atkinson’s definition (but is stressed by others, e.g., Yaden et al 2017, Lindstrom et al 2022) and allows us to analyze it alongside other items that also involve alterations in the sense of self. This phenomenologically oriented analysis allowed us to suggest why we were able to validate some of the Sense of Self items and not others, which was the primary aim of our discussion. Our decision to approach the Absorbed item in this way was reenforced by Lindstrom et al (2022), an article recommended by R2 and discussed below.

I have found a few minor flaws:

1. Table 1, “ANNEX”: perhaps AANEX? Yes, corrected.

2. P. 16, l. 6, Mental state: It seems to me that it could be implemented with the item “Other”; in fact, these kinds of experiences may be also favored by a variety of factors non mentioned in the text, such as psychotropic drugs. “Using drugs or alcohol” is one of the response options; respondents interpreted it to include psychotropic drugs (as we intended). Attempts to differentiate between different categories of drugs did not yield cross-culturally stable distinctions. We opted for “None of the Above” instead of an “Other” response.

3. TP. 29, l. 10, the formula and the following similar formulas are not clear to me (I only see squares), and I am afraid it will not be clear to the reader as well. We redid the formulas.

P. 44, l. 1, “mediation”: perhaps “meditation”? Yes, corrected.

Reviewer #2: We appreciate R2’s positive response to the paper and the minor changes s/he suggests. Our responses are indicated below: 

Additional Limitations: Although the small N is mentioned as a limitation, there are additional limitations that should be mentioned: 

a) most respondents are male, despite the fact that typically females endorse more most of these experiences. 

b) although the procedure makes it unlikely that respondents were robots or non-attentive, nonetheless the problems with Mechanical Turk should be mentioned, e.g., Perspectives on Psychological Science: "Too Good to Be True: Bots and Bad Data From Mechanical Turk." by Margaret A. Webb and June P. Tangney. We expanded the limitations section to address representativeness and participant quality on MTurk.

We used MTurk workers as a proxy for the general population in each country, recognizing that neither sample is completely representative of the population of their country. For example, MTurk users in the United States have been shown to skew male, as in the current study, as well as to score higher on measures of negative affect and social anxiety (Arditte et al., 2016; McCredie et al., 2019). MTurk also has well-known issues with data quality (Moss & Litman, 2020; Webb & Tangney, 2022), even relative to other online survey platforms (Douglas et al., 2023). In light of these limitations, we recommend that researchers consider alternative online survey platforms, such as Prolific, which may yield higher-quality data.

c) it is VERY good that a question about mental states was added, but the list of possibilities is small and does not contain, for instance, meditation. We recognize that there are a variety of other factors, including religious or spiritual practices, such as meditation, that are not specifically mentioned and thus by default fall under the heading of “None of the Above.” As indicated in the results section (p. 44), we attempted to validate a practice-related response option but were unable to do so without getting culture-specific. 

Literature Review: The literature review is fairly comprehensive. I would nonetheless recommend these two items: 1) Acta Psychiatrica Scandinavica, Evidence for the early clinical relevance of hallucinatory-delusional states in the general population, R. Nuevo, J. Van Os, C. Arango, S. Chatterji, J. L. Ayuso-Mateos, for cross-cultural differences in a type of unusual experience (hallucination), 2) Dissolution of What? The Self Lost in Self-Transcendent Experiences. Lindström, L., Kajonius, P. & Cardeña, E., 2022, In: Journal of Consciousness Studies. 29, 5-6, p. 75-101, for an example of teasing out the conceptual vagueness of the term "ego dissolution." We couldn’t figure out a place where the first article was relevant, but were very excited to learn of the second article. We are citing it in this paragraph and making some minor edits.

The analysis of the four Sense of Self items that we were able to validate (discussed here) and the nine we were not (discussed in the next section) suggests that some types of changes in sense of self may be more amenable to cross-cultural survey study than others. In discussing our ability to validate the Sense of Self items, we can distinguish between items that refer to the disruption of high-level reflective processing (the narrative self) and those that refer to lower-level processes (e.g., “minimal” or “embodied” self; for reviews, see Johnstone et al., 2021; Lindström et al., 2022; Millière, 2017). The former includes those items that involve a decrease in self-related thought, the loss of access to semantic autobiographical information, and discontinuities in the sense of personal identity. The latter includes changes in self-location, body ownership, or body awareness (for a discussion, see Lindström et al., 2022; Millière et al., 2018). 

We were impressed by the results Lindholm et al 2022 reported based on in-depth interviews of a targeted sample, so added a sentence in the discussion of the mystical experience-related items acknowledging the advantages of interviews over surveys in investigating experiences that involved a profound sense of self loss.

Our inability to validate these items suggests that the scales from which these items were drawn capture a much wider range of experiences than they intend and may, thus, be an ineffective way to measure the shift in the sense of self that, according to some Western traditions, is the hallmark of a mystical experience. It may be that there is a common phenomenological feature that can be characterized as an experience of “undifferentiated unity” or “ego dissolution” that could be captured by improved survey items. It is possible that the experience is too subtle to characterize in the everyday language of the lay population and that a better formulation would require the use of more specialized vocabularies better suited to describing such experiences. Or it may be that the experience of “undifferentiated unity” does not refer to a specific feature. Carhart-Harris, who was involved in creating the Ego Dissolution Inventory (Nour et al., 2016), has cautioned researchers that “‘self-loss’ and related expressions such as ‘ego dissolution’ are notoriously ambiguous notions” and recommended the multidimensional approach to specifying disruptions in sense of self adopted here (Millière et al., 2018). Lindström et al. (2022) share this concern and demonstrate the value of in-depth interviews of subjects who report experiences that involve a profound sense of self loss in teasing apart the different aspects of such experiences.

Regardless of whether it is possible to validate a more refined “Unity” or “Ego Dissolution” item for use in surveys of the general population, it should be noted that the mysticism construct, as defined by Stace and operationalized in the mysticism scales … 

Mysticism Discussion: R2 writes: It is mentioned on page 60 that the mysticism construct, as defined by Stace and operationalized in the mysticism scales, “presupposes, on theological grounds, that experiences of undifferentiated unity are inherently positive, that positive valence is part of the experience itself, and that the experience is culturally unmediated (i.e., that the cultural environment does not play a significant role in constituting the positively valenced experience as it unfolds)” (76), but that is not the conclusion reached by Wulff (reference 71), that mentioned that such experiences are not necessarily positive and that culture PARTLY mediates the experience. This should be mentioned, as well as his (and others}) categorization of different types of mystical experiences, which may explain part of the problem with the question used. We recognize that Wulff takes a broad approach and recognizes the range of experiences that have been characterized as mystical. That is why his chapter is cited (p. 58) as offering an overview of the diversity of experiences characterized as such. The quoted statement refers specifically to the construct defined by Stace and operationalized in the mysticism scales (M-Scale and MEQ), from which we drew items that we attempted to validate. We query other phenomenological features sometime associated with mystical experiences, e.g., peace, joy, and bliss, in two separate items (Peace and Joy). 

Another Self in Body item: Given the small and technologically savvy sample, the problem [with this item] may be that no one was involved in ritual possessions and thus could not attest to this. And even in ritual possessions there are differences in, e.g., amount of amnesia, see Cardeña, E., Schaffler, Y., & Van Duijl, M. (2023). The other in the self: Possession, trance, and related phenomena. In M. J. Dorahy, S. N. Gold, & J. A. O’Neill (Eds.), Dissociation and the dissociative disorders: Past, present, future. 2nd ed. (pp. 421-432). Routledge. doi: 10.4324/9781003057314-31. This is a helpful reference that is consistent with our finding of inconsistency in reported experiences. We have clarified the relevant paragraph to stress that we were surveying a general population in which there were few such experiences and that the literature, including the suggested citation, suggest that in a larger targeted sample the phenomenology of experiences characterized as possession would not be consistent.

In the case of Another Self in Body, which attempted to capture the most marked disruptions in the narrative self, the item was well understood by those who did not endorse it, many of whom gave possession as an example of such an experience. However, it was not well understood by those who did endorse it. Of the few (6/45) who claimed to have had such an experience, most described a co-conscious, conscience-like “inner voice” and/or a sense of inner conflict or being “of two minds.” Moreover, the INOE as a self-report measure can only capture experiences in which subjects are co-conscious with (rather than displaced by) the alien self. This suggests that, while the cultural concept of another (alien) self in the body is widely understood in the general population, the subjective experience of the few who claim to have had the experience is not consistent. The work of ethnographers and historians, however, would suggest that even in a larger sample of people whose behaviors were locally characterized as “spirit possession” (Cardena et al 2023; Harvey, 2010; Humphrey, 1994) or “avesta” (Sanskrit/Hindi: Malhotra, 2022; Smith, 2006) we would likely have found a variety of subjective sensations, which they or those around them appraise as self-alien. Moreover the sense of switching between “selves” can be present in various contexts where it is understood in different ways, e.g., when bicultural individuals switch cultural frames (Ramírez-Esparza et al., 2006) or actors perform in theaters (Panero et al., 2020). 

5) The references list needs to be proofed better. For instance, capitalization of paper titles is quite inconsistent. References were proofread and capitalization was standardized following the PLOS One reference guidelines.

---

## [Editor Report · Decision Letter 1]

13 Jun 2023

The Inventory of Nonordinary Experiences (INOE): Evidence of validity in the United States and India

PONE-D-23-05816R1

Dear Dr. Taves,

We’re pleased to inform you that your manuscript has been judged scientifically suitable for publication and will be formally accepted for publication once it meets all outstanding technical requirements.

Kind regards,

Luca Valera

Academic Editor

PLOS ONE
---

## [Editor Report · Acceptance letter]

4 Jul 2023

PONE-D-23-05816R1 

The Inventory of Nonordinary Experiences (INOE): Evidence of validity in the United States and India 

Dear Dr. Taves:

I'm pleased to inform you that your manuscript has been deemed suitable for publication in PLOS ONE. Congratulations! Your manuscript is now with our production department. 

Kind regards, 

on behalf of

Dr. Luca Valera 

Academic Editor

PLOS ONE